# Integration of sensory and fear memories in the rat medial temporal lobe

Francesca S Wong, Alina B Thomas, Simon Killcross, Vincent Laurent, R Fred Westbrook, Nathan M Holmes*

School of Psychology, University of New South Wales, Sydney, Australia

## eLife Assessment

This **important** study by Wong et al. addresses a long-standing question in the field of associative learning regarding how a motivationally relevant event can be inferred from prior learning based on neutral stimulus-stimulus associations. The research provides **convincing** behavioral and neurophysiological evidence to address this question. The article will be interesting for researchers in behavioral and cognitive neuroscience.

**Abstract** Wong et al., 2019 used a sensory preconditioning protocol to examine how sensory and fear memories are integrated in the rat medial temporal lobe. In this protocol, rats integrate a sound-light (sensory) memory that forms in stage 1 with a light-shock (fear) memory that forms in stage 2 to generate fear responses (freezing) across test presentations of the sound in stage 3. Here, we advance this research by showing that (1) how/when rats integrate the sound-light and light-shock memories (online in stage 2 or at test in stage 3) changes with the number of sound-light pairings in stage 1; and (2) regardless of how/when it occurs, the integration requires communication between two regions of the medial temporal lobe: the perirhinal cortex and basolateral amygdala complex. Thus, 'event familiarity' determines how/when sensory and fear memories are integrated but not the circuitry by which the integration occurs: this remains the same.

## Introduction

Memory is the means by which the past connects with the present to guide our future interactions with the environment. One of the ways it achieves this feat is by integrating the sensory and emotional elements of experiences that are separated in time (e.g., experiences that occur days, weeks, or months apart). This allows people and animals to better anticipate and react to different types of situations. For example, after being assaulted by person A, you may be apprehensive when you encounter their known-associate, person B, who was not present at the time of the assault; after being attacked by a dog in the street, a child may avoid places where the dog had been previously encountered (e.g., the park); and after learning the relationship between a particular smell and danger, an animal may avoid places where it had previously encountered that smell.

How does the brain integrate sensory and emotional memories so that animals/people can anticipate and react to different types of situations? This question can be addressed through the study of sensory preconditioned fear in rats. A standard protocol to produce sensory preconditioned fear consists in three stages. In stage 1, rats are exposed to pairings of two relatively innocuous stimuli, for example, a sound followed by a light. In stage 2 (typically 24 h later), one of these stimuli, for example, the light, is paired with danger in the form of brief but aversive foot shock. Finally, in stage 3, rats exhibit fear responses, for example, suppression of food-reinforced lever pressing or freezing, when tested with the light that had been paired with danger or the sound that had never paired with

*For correspondence:
n.holmes@unsw.edu.au

danger. Importantly, fear of the sound is not due to generalization from the conditioned light, as controls exposed to explicitly unpaired presentations of the relevant stimuli in either stage of training do not show this fear (*Fam et al., 2023*; *Holmes et al., 2013*; *Holmes and Westbrook, 2017*; *Kikas et al., 2021*; *Michalscheck et al., 2021*; *Rizley and Rescorla, 1972*; *Wong et al., 2019*). Instead, fear of the sound is due to integration of memories that form in each stage of training: the sensory sound-light memory that forms in stage 1 is integrated with the emotional light-shock memory that forms in stage 2 to generate fear of the sound when it is tested in stage 3.

When are the two memories integrated? In a previous study published in *eLife*, *Wong et al., 2019* showed that integration of the sound-light and light-shock memories occurs 'online' at the time of the light-shock pairings in stage 2. This conclusion was based on a series of experiments which examined the involvement of the perirhinal cortex (PRh) in stage 2 of the protocol described above. The PRh was selected as the region of interest as it encodes the sound-light memory that forms in stage 1 (*Holmes et al., 2013*; *Holmes et al., 2018*; *Qureshi et al., 2023*) but plays no role in the formation of the directly conditioned light-shock memory in stage 2 (*Bang and Brown, 2009*; *Campeau and Davis,*

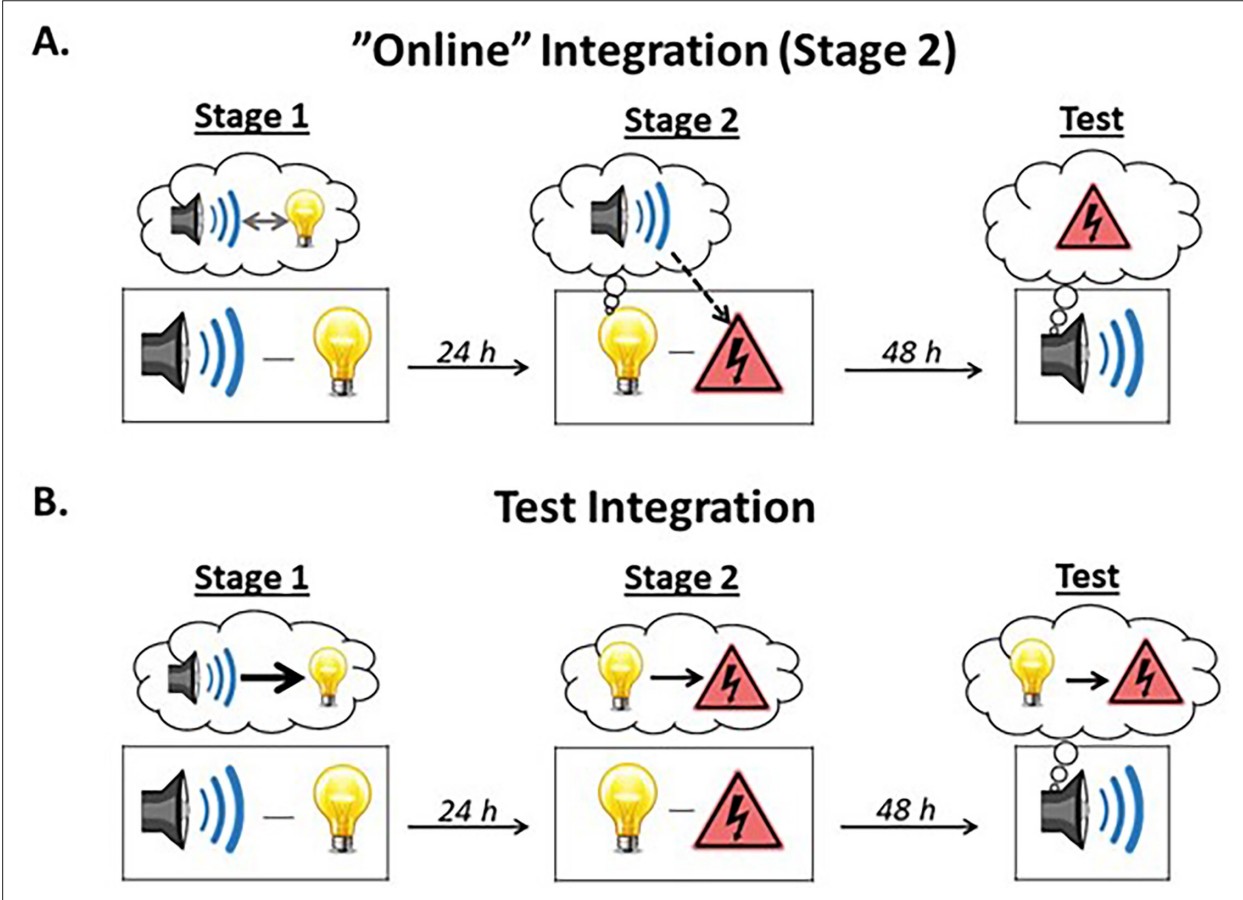

**Figure 1.** Illustration of the two types of integration. (**A**) The sound-light and light-shock memories can be integrated 'online' during stage 2. When the subject is exposed to light-shock pairings in stage 2, the light activates the memory of its past associate, the sound, thereby allowing it to associate with the shock (i.e., subjects form a mediated sound-shock association). Test presentations of the sound then retrieve this mediated sound-shock association, resulting in the expression of fear. This type of integration is hypothesized to occur after few sound-light pairings in stage 1. Under these circumstances, subjects do not encode the order in which the events occur, resulting in mediated learning about the sound across direct conditioning of the light. (**B**) Alternatively, the sound-light and light-shock memories can be integrated at the time of testing. Here, when the sound is presented at test, the subject retrieves the sound-light memory formed in stage 1 and integrates (or chains) it with the light-shock memory formed in stage 2, resulting in the expression of fear. This type of integration is hypothesized to occur after many sound-light pairings in stage 1. Under these circumstances, subjects learn that the sound is followed by the light (sound →light) and that that light is followed by nothing (light→nothing). Hence, during the session of light-shock pairings in stage 2, the light does *not* activate the memory of the sound and the mediated sound-shock association does not form. Instead, during testing with the sound in stage 3, the light and its shock associate are strongly called to mind via the chain, sound→light→shock, resulting in the expression of fear responses.

*1995*; *Kholodar-Smith et al., 2008*; *Lindquist et al., 2004*; *Romanski and LeDoux, 1992*; *Wilensky et al., 2006*). Indeed, Wong et al. confirmed past findings that neuronal activity in the PRh is *not* needed for acquisition of directly conditioned fear to the light. By contrast, the same testing revealed that neuronal activity in the PRh *is* needed for acquisition of sensory preconditioned fear to the sound. Wong et al reasoned that, if fear of the sound was due to chaining of sound-light and light-shock memories at the time of testing (the alternative hypothesis; *Figure 1*), it should have been intact as the two component memories were unaffected by the PRh manipulations. That is, as the manipulations were administered days after the session of sound-light pairings, the sound-light memory will have been intact; and the manipulations had no effect on acquisition of fear to the light showing that the light-shock memory was also intact (see also *Holmes et al., 2022a*). Thus, the finding that fear of the sound was abolished shows that, when it occurs, it must be supported by some 'online' process across the session of light-shock pairings. Specifically, the results were taken to imply that, when the light is paired with shock in stage 2, it activates a PRh-dependent memory of its past associate, the sound, resulting in a direct light-shock memory as well as an indirect or mediated sound-shock memory (*Holland, 1981*; *Holmes et al., 2022b*; *Lingawi et al., 2018*; *Wong et al., 2019*). Hence, when either stimulus is presented at test, rats expect the shock and freeze in its anticipation.

The findings reported by *Wong et al., 2019* raise two questions. The first is whether integration of the sound-light and light-shock memories *always* occurs online across conditioning of the light. While Wong et al. provided evidence for such online integration, under some circumstances it may become more dependent on chaining of the sound-light and light-shock memories at test. One of the ways in which this could occur is by increasing the number of sound-light pairings from that used by Wong et al. (*Holland, 1998*; *Holmes et al., 2022b*). Such an increase could alter what is encoded across preconditioning from a bidirectional sound↔light associative structure to one which just encodes the order in which the events were presented; namely, a unidirectional sound→light structure. The shift from the former to the latter with an increase in the number of sound-light pairings would thus undermine the ability of the light to activate the memory of the sound across conditioning and, hence, mediate conditioning of the sound. At the same time, such an increase would strengthen the ability of the sound to activate the representation of the now-conditioned light at test, initiating the memorial chain that enables the sound to elicit freezing (see *Figure 1*). That is, increasing the number of sound-light pairings may allow rats to encode information about stimulus order in stage 1 and, thereby, shift the locus of integration from mediated conditioning in stage 2 to chaining at test in stage 3 (*Holmes et al., 2022a*).

The second question raised by the *Wong et al., 2019* findings concerns the neural substrates underlying integration. Regardless of when it occurs, integration must involve interactions between the basolateral amygdala complex (BLA) and the PRh. Neuronal activity in the BLA, but not the PRh, is critical for the formation of the light-shock memory and its expression in fear responses (e.g., *Maren et al., 1996*; *Schafe et al., 2000*; see also *Duvarci and Pare, 2014*; *Rodrigues et al., 2004* for reviews); whereas neuronal activity in both the PRh (*Holmes et al., 2013*; *Wong et al., 2019*) and BLA (*Holmes et al., 2013*) is required for the expression of fear responses to the sensory preconditioned sound. These findings suggest that communication between the PRh and BLA may be required for successful integration of the memories formed in training and, thereby, expression of fear to the sound. However, it remains to be determined whether this is, in fact, the case. That is, it remains to be determined whether communication between the PRh and BLA is required for 'online' integration of the sound-light and light-shock memories in stage 2 and/or completion of a sound→light→shock memory chain at test in stage 3.

The present study addressed both of the questions raised by the *Wong et al., 2019* findings. It had two specific aims. The first aim was to determine whether the type of integration depends on the number of sound-light pairings in stage 1: that is, whether the sound-light and light-shock memories are integrated 'online' via mediated conditioning after few sound-light pairings, and via chaining at the time of testing after many sound-light pairings. We addressed this aim by examining whether sensory preconditioned fear of the sound requires activation of *N*-methyl-D-aspartate (NMDA) receptors in the PRh across the session of light-shock pairings in stage 2. If integration occurs via mediated conditioning, as we have shown in the case of rats exposed to few sound-light pairings in stage 1, then blocking NMDA receptors in the PRh (via an infusion of the NMDA receptor antagonist, DAP5) immediately prior to the session of light-shock pairings should disrupt fear of the sound at test (measured

in freezing). If, however, integration occurs through chaining of the sound-light and light-shock memories at test, as we expect in the case of rats exposed to many sound-light pairings in stage 1, blocking NMDA receptors in the PRh in stage 2 should have no effect on fear of the sound at test.

The second aim of this study was to determine whether integration of the sound-light and light-shock memories depends on communication between the PRh and BLA: that is, whether this communication is required for mediated conditioning of the sound across the light-shock pairings in stage 2, and/or chaining of the sound-light and light-shock memories across testing of the sound in stage 3. We addressed this aim by functionally disconnecting the PRh and BLA immediately prior to the session of light-shock pairings in stage 2 or testing with the sound in stage 3. If communication between the PRh and BLA is required for mediated conditioning of the sound, disconnecting the PRh and BLA prior to the session of light-shock pairings should disrupt fear of the sound at test; and if communication between the PRh and BLA is required for completion of the sound→light→shock memory chain, disconnecting the PRh and BLA before the test session should again disrupt fear of the sound.

## Results

### Experiment 1: A demonstration of sensory preconditioned fear after few or many sound-light pairings in stage 1

The aim of Experiment 1 was to demonstrate sensory preconditioned fear among rats exposed to either the same number (8) of sound-light pairings used by *Wong et al., 2019* or four times as many (32) pairings. Four groups of rats were exposed to presentations of a sound and light in stage 1. Two of these groups were exposed to just 8 presentations of each stimulus (Groups P8 and U8), while the remaining two groups were exposed to 32 presentations of each stimulus (Groups P32 and U32). For one group in each of these pairs, the sound and light were paired (P) such that every sound presentation was immediately followed by a presentation of the light (Groups P8 and P32); while, for the remaining group, the sound and light were unpaired (U) such that each sound and light presentation occurred several minutes apart (Groups U8 and U32). All rats were then exposed to light-shock pairings in stage 2 (which commenced 24 h after stage 1) and, finally, tested with presentations of the sound alone and light alone in stage 3 (which commenced 48–72 h after stage 2).

#### Conditioning

During conditioning in stage 2 and testing in stage 3, the baseline levels of freezing were low (<10%) and did not differ between the groups (largest $F < 3.993$, p>0.05). Conditioning of the light in stage 2 was successful in all groups. The mean (± SEM) levels of freezing to the light on its final pairing with shock were 72.5 ± 9.2% in Group P8, 88.6 ± 6.0% in Group U8, 67.5 ± 13.6% in Group P32, and 82.9 ± 11.9% in Group U32. There was a significant linear increase in freezing across the light-shock pairings ($F_{(1,26)} = 158.68$; p<0.001; $n_p^2 = 0.859$, 95% CI: [1.929, 2.681]). The rate of increase did not differ between groups that had been exposed to either 8 or 32 sound and light presentations in stage 1 ($F_{(1,26)} = 1.306$; p=0.264); nor between groups that had been exposed to paired or unpaired sound and light presentations in stage 1 ($F_{(1,26)} = 1.852$; p=0.185). There was no interaction between the rate of increase, the number of sound and light presentations, and the relation between the sound and light ($F_{(1,26)} = 0.283$; p=0.599). The overall comparisons of freezing to the light (averaged across the four conditioning trials) between groups that received either paired or unpaired stimulus presentations in stage 1 (factor 1), and between groups that received either 8 or 32 sound and light exposures in stage 1 (factor 2), were not significant ($F$s < 0.45, p>0.508). The interaction between these two between-subject factors was also not significant ($F < 0.45$, p>0.508).

#### Test

*Figure 2B* shows the mean (± SEM) levels of freezing to the sound alone (left panel) and light alone (right panel) averaged across the eight test presentations of each stimulus in Experiment 1. Rats that had been exposed to 32 presentations of the sound and light in stage 1 froze significantly more to the sound than rats that had been exposed to just eight presentations of the sound and light ($F_{(1,26)} = 38.986$; p<0.001; $n_p^2 = 0.600$; 95% CI: [0.757, 1.500]); and rats that had been exposed to sound-light *pairings* in stage 1 froze significantly more to the sound than rats that had been exposed to *unpaired* presentations of the sound and light ($F_{(1,26)} = 75.983$; p<0.001; $n_p^2 = 0.745$; 95% CI: [1.204, 1.947]).

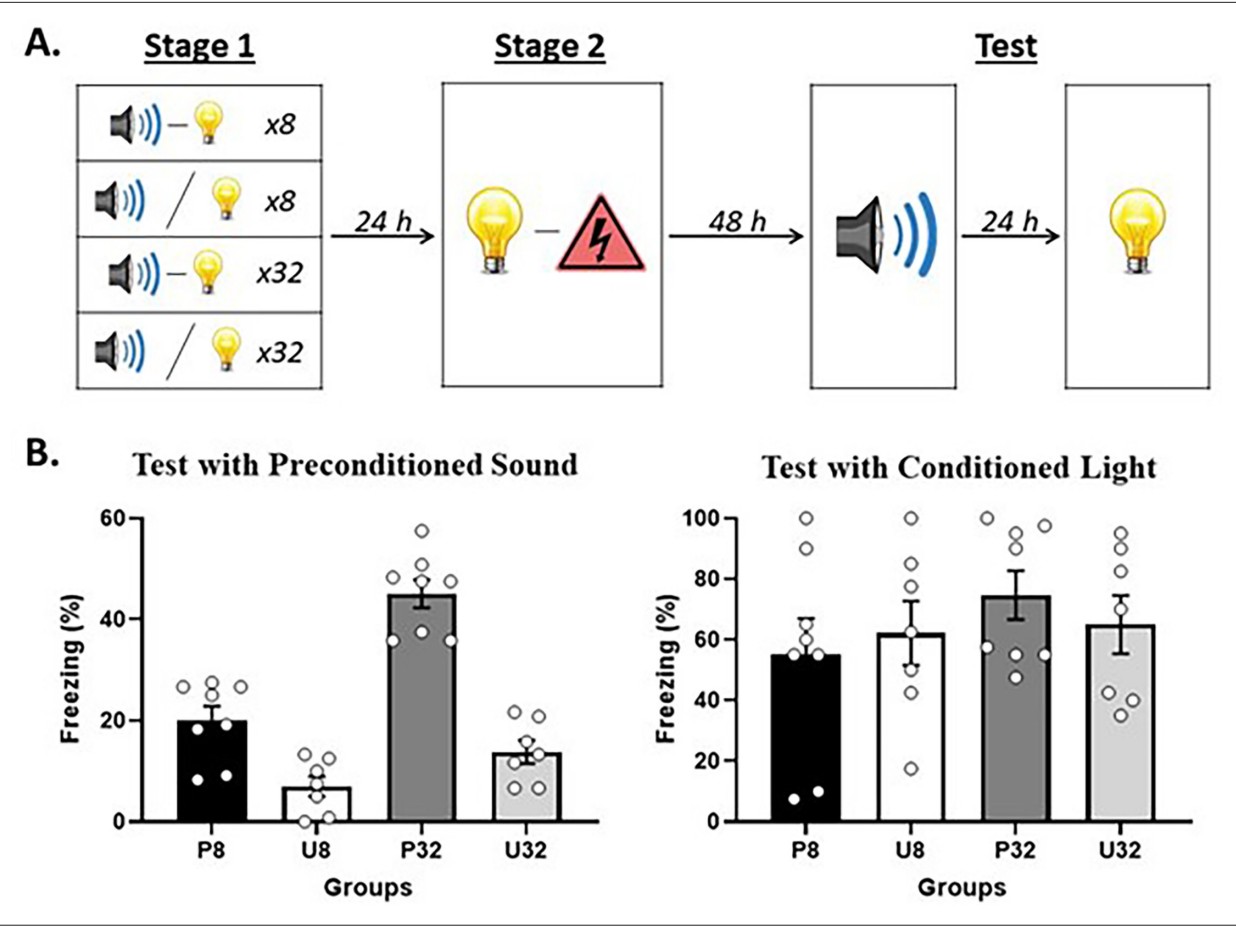

**Figure 2.** Demonstration of sensory preconditioned fear after few or many sound-light pairings in stage 1. (**A**) Schematic of the behavioral protocol. In stage 1, rats were exposed to presentations of a sound and light. They either received 8 paired presentations in Group P8 (n = 8), 8 unpaired presentations in Group U8 (n = 7), 32 paired presentations in Group P32 (n = 8), or 32 unpaired presentations in Group U32 (n = 7). In stage 2, all rats received paired presentations of the light and shock. Finally, in stage 3 (Test), all rats were tested with the preconditioned sound alone and then with the conditioned light alone. (**B**) Percentage freezing to the preconditioned sound (left panel) and to the conditioned light (right panel), averaged across the eight trials of their respective tests. Data shown are means ± SEM.

Importantly, there was a significant interaction between the number of stimulus presentations in stage 1 and the relation between the two stimuli ($F_{(1,26)}$ = 12.802; p=0.001; $n_p^2$ = 0.330; 95% CI: [0.275, 1.018]). Inspection of *Figure 2B* shows that this was due to the difference in freezing to the sound between Groups P32 and U32 being greater than the difference in freezing to the sound between Groups P8 and U8. Together, these and our previous findings indicate that freezing to the sound requires formation of sound-light and light-shock memories in training (i.e., it is not due to generalization), and does not decrease as the number of sound-light pairings in stage 1 is increased: in fact, the evidence for sensory preconditioned fear increased with the number of sound-light pairings. Finally, there were no significant differences between the groups in levels of freezing to the light alone (*F*s < 1.21; p>0.281).

## Experiments 2A and 2B: The two types of integration can be dissociated by a stage 2 infusion of the NMDA receptor antagonist, DAP5

The sensory memory produced by pairings of an auditory and visual stimulus (e.g., sound-light) requires activation of NMDA receptors in the PRh (*Holmes et al., 2013*), whereas the fear memory produced by pairings of an auditory or visual stimulus with foot shock (e.g., light-shock) requires activation of NMDA receptors in the BLA (*Bauer et al., 2002*; *Keidar et al., 2023*; *Rodrigues et al.,*

*2001*; *Williams-Spooner et al., 2022*). Accordingly, Experiments 2A and 2B examined whether sensory preconditioned fear of a sound requires activation of PRh NMDA receptors during the stage 2 session of light-shock pairings; and, critically, whether any such requirement depends on the number of sound-light pairings in stage 1.

In Experiment 2A, two groups of rats were exposed to eight sound-light pairings in stage 1 and a session of light-shock pairings in stage 2. Immediately prior to the stage 2 session, rats in one group received a PRh infusion of the NMDA receptor antagonist, DAP5, while rats in the other group received a PRh infusion of vehicle only. All rats were then tested with presentations of the sound alone and light alone in stage 3. Experiment 2B also involved two groups of rats. The treatment of these groups differed from their Experiment 2A counterparts in just one respect: in stage 1, they were exposed to 32 sound-light pairings.

*Wong et al., 2019* showed that, among rats exposed to eight sound-light pairings in stage 1, acquisition of fear to the sound requires neuronal activity in the PRh in stage 2. Accordingly, among rats that received this number of sound-light pairings (Experiment 2A), we expected that acquisition of fear to the sound would require activation of PRh NMDA receptors in stage 2 and, hence, be impaired by the PRh infusion of DAP5 in stage 2. Such results would add further support to the claim that integration of the sound-light and light-shock memories in the protocol used by Wong et al. occurs through formation of a mediated sound-shock memory in stage 2, at the time of conditioning to the light.

By contrast, we expected the effects of the PRh DAP5 infusion to be different among rats exposed to 32 sound-light pairings in stage 1 (Experiment 2B). Specifically, we hypothesized that increasing the number of sound-light pairings from 8 to 32 would change the way that rats integrate the sound-light and light-shock memories: they would cease to be integrated through formation of a mediated sound-shock memory and, instead, be integrated through chaining at the time of testing with the sound. Hence, we expected that, among rats exposed to 32 sound-light pairings in Experiment 2B, acquisition of fear to the sound would *not* require activation of PRh NMDA receptors in stage 2 and, hence, be unaffected by the PRh infusion of DAP5 immediately prior to stage 2: that is, rats that receive a PRh infusion of DAP5 in stage 2 would freeze just as much during test presentations of the sound alone as the vehicle-infused controls.

## Experiment 2A: Rats exposed to eight sound-light pairings in stage 1

### Conditioning

The baseline levels of freezing during conditioning and test sessions were low (<10%) and did not differ between groups (largest $F_{(1,16)}$ = 2.234; p=0.154). Conditioning of the light was successful. The mean (± SEM) levels of freezing to the light on its final pairing with shock were 84.4 ± 10.9% in Group 8-VEH and 88.9 ± 7.5% in Group 8-DAP5. Freezing to the light increased across the four light-shock pairings in stage 2 ($F_{(1,16)}$ = 70.89; p<0.001; $n_p^2$ = 0.816; 95% CI: [1.656, 2.771]). The rate of this increase did not differ between groups ($F_{(1,16)}$ = 0.339; p=0.569), and there were no significant between-group differences in overall freezing to the light ($F_{(1,16)}$ = 0.248; p=0.625).

### Test

*Figure 3B* shows mean (± SEM) levels of freezing averaged across the eight test presentations of the sound alone (left panel) and light alone (right panel) in Experiment 2A. Both groups showed equivalent levels of freezing to the directly conditioned light ($F_{(1,16)}$ < 0.248; p=0.625). However, rats that received a PRh infusion of DAP5 (Group 8-DAP5) in stage 2 exhibited significantly less freezing to the preconditioned sound than rats that received a PRh infusion of vehicle only (Group 8-VEH; $F_{(1,16)}$ = 18.645; p=0.001; $n_p^2$ = 0.538; 95% CI: [0.622, 1.821]).

## Experiment 2B: Rats exposed to 32 sound-light pairings in stage 1

### Conditioning

The baseline levels of freezing during conditioning and test sessions were low (<10%) and did not differ between groups (largest $F_{(1,17)}$ = 3.099; p=0.096). The mean (± SEM) levels of freezing to the light on its final pairing with shock were 95.6 ± 2.9% in Group 32-VEH and 92 ± 6.1% in Group 32-DAP5. Conditioning of the light was successful. Freezing to the light increased across the four light-shock pairings ($F_{(1,17)}$ = 345.48; p<0.001; $n_p^2$ = 0.953; 95% CI: [2.403, 3.018]). The rate of this increase did not

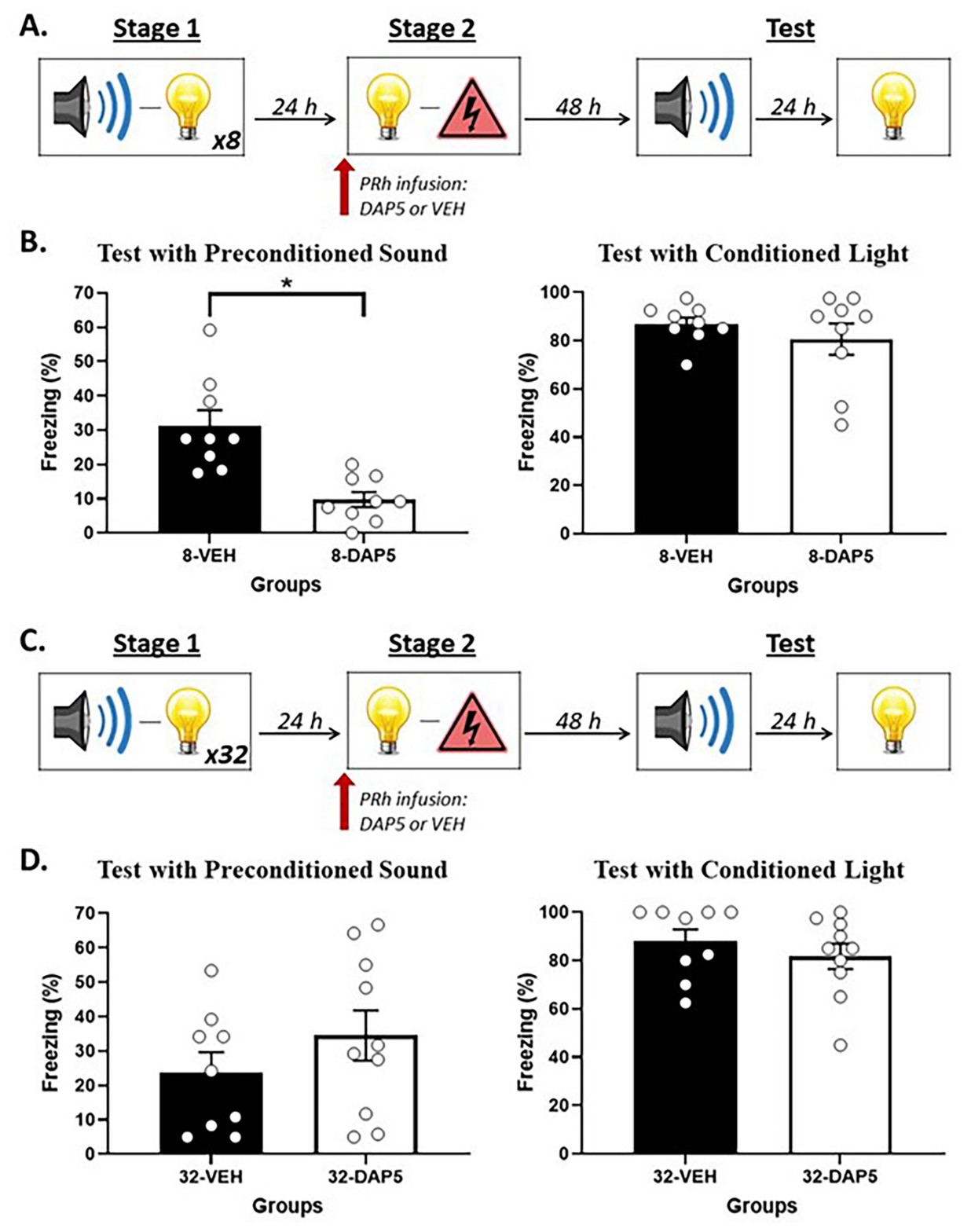

**Figure 3.** *N*-methyl-ᴅ-aspartate (NMDA) receptor activity in the perirhinal cortex (PRh) supports online integration in stage 2 after few sound-light pairings in stage 1 but NOT after many sound-light pairings. (**A**) Schematic of the behavioral procedure for Experiment 2A (Group 8-VEH, n = 9; and Group 8-DAP5, n = 9). The red arrow indicates that the infusion of either DAP5 or vehicle (VEH) occurred before stage 2. (**B**) Percentage freezing to the preconditioned sound (left panel) and to the conditioned light (right panel), averaged across the eight trials of their respective tests. Data shown are means ± SEM. The asterisk (*) denotes a statistically significant difference (p<0.05). (**C**) Schematic of the behavioral procedure for Experiment 2B (Group

*Figure 3 continued*

32-VEH, n = 9; and Group 32-DAP5, n = 10). The red arrow indicates that the infusion of either DAP5 or VEH occurred before stage 2. (**D**) Percentage freezing to the preconditioned sound (left panel) and to the conditioned light (right panel), averaged across the eight trials of their respective tests. Data shown are means ± SEM.

differ between groups ($F_{(1,17)}$ = 0.197; p=0.663), and there were no significant between-group differences in overall freezing to the light ($F_{(1,17)}$ = 0.650; p=0.431).

### Test

*Figure 3D* shows mean (± SEM) levels of freezing averaged across the eight test presentations of the sound alone (left panel) and light alone (right panel) in Experiment 2B. The statistical analysis confirmed what is evident in the figure: both groups showed equivalent levels of freezing to the directly conditioned light ($F_{(1,17)}$ < 0.752; p=0.398), and to the sensory preconditioned sound ($F_{(1,17)}$ = 1.270; p=.275).

This experiment has replicated previous findings that the PRh is not involved in the conditioning of fear responses to a stimulus that is directly paired with foot shock (*Bang and Brown, 2009*; *Campeau and Davis, 1995*; *Kholodar-Smith et al., 2008*; *Lindquist et al., 2004*; *Romanski and LeDoux, 1992*; *Wilensky et al., 2006*; *Wong et al., 2019*). It has also revealed two new findings. The first is that, after few (eight) sound-light pairings in stage 1, acquisition of fear to the sound requires activation of PRh NMDA receptors across the session of light-shock pairings in stage 2 (Experiment 2A). The second is that, after many (32) sound-light pairings in stage 1, acquisition of fear to the sound does not require activation of PRh NMDA receptors in stage 2. Taken together, these results are consistent with the hypothesis that increasing the number of sound-light pairings changes the way that rats integrate the sound-light and light-shock memories in a sensory preconditioning protocol. After few sound-light pairings, rats form a mediated sound-shock memory during conditioning of the light. After many sound-light pairings, rats no longer form a mediated sound-shock memory but, instead, chain the sound-light and light-shock memories when tested with the sound.

## Experiments 3A and 3B: After a few sound-light pairings, integration requires communication between the PRh and BLA in stage 2; after many sound-light pairings, it does not

The PRh and BLA share strong reciprocal connections (*Höistad and Barbas, 2008*; *McDonald, 1998*; *McIntyre et al., 1996*; *Ottersen, 1982*; *Pitkänen et al., 2000*; *Shi and Cassell, 1999*) and work together to support different aspects of memory in rats (*Gómez-Chacón et al., 2012*; *Perugini et al., 2012*; *Santos et al., 2023*), cats (*Bauer et al., 2007*; *Collins et al., 2001*; *Paz et al., 2006*), monkeys (*Mogami and Tanaka, 2006*; *Ohyama et al., 2012*), and people (*Cooper et al., 2023*; *Dolcos et al., 2004*; *Mochizuki et al., 2020*; *Ritchey et al., 2008*). Here, we examined whether communication between the PRh and BLA is necessary for integration of the sound-light and light-shock memories that form in sensory preconditioning. We first examined this communication requirement in stage 2 of the protocols involving either few (8; Experiment 3A) or many (32; Experiment 3B) sound-light pairings in stage 1. In each of these experiments, rats underwent surgery in which an excitotoxic lesion of the PRh was created in one hemisphere and a single cannula was implanted in the BLA in either the same hemisphere (i.e., ipsilateral to the lesion; Groups IPSI) or the opposite hemisphere (i.e., contralateral to the lesion; Groups CONTRA). After recovery, these rats were exposed to either 8 (Experiment 3A) or 32 (Experiment 3B) sound-light pairings in stage 1, and a session of light-shock pairings in stage 2. Immediately prior to this session, rats received a BLA infusion of the GABA$_A$ receptor agonist, muscimol, which is widely used to silence neuronal activity (e.g., *Helmstetter and Bellgowan, 1994*; *Wilensky et al., 2006*; *Holmes et al., 2013*). This infusion transiently disconnects the PRh and BLA for rats in Groups CONTRA as these rats do not have a functioning PRh *and* BLA in either hemisphere, but spares PRh-BLA connectivity for rats in Groups IPSI, as these rats have a functioning PRh and BLA in one hemisphere. Finally, all rats were tested with presentations of the sound alone and light alone in stage 3.

The previous experiment showed that increasing the number of sound-light pairings in stage 1 changes how rats integrate the sound-light and light-shock memories to generate fear of the sound:

it functionally shifts the integration from mediated (sound-shock) learning in stage 2 to chaining at test in stage 3. Accordingly, we hypothesized that, among rats exposed to eight sound-light pairings in stage 1 (Experiment 3A), integration of the sound-light and light-shock memories would require communication between the PRh and BLA in stage 2; and, hence, be impaired by the PRh-BLA disconnection in Group 8-CONTRA. That is, relative to rats in Group 8-IPSI, we expected that rats in Group 8-CONTRA would freeze just as much when tested with presentations of the directly conditioned light but significantly less when tested with presentations of the preconditioned sound. By contrast, among rats exposed to 32 sound-light pairings in stage 1 (Experiment 3B), we hypothesized that fear of the sound would not require communication between the PRh and BLA in stage 2; and, hence, would not be impaired by the PRh-BLA disconnection in Group 32-CONTRA. That is, relative to rats in Group 32-IPSI, we expected that rats in Group 32-CONTRA would freeze just as much when tested with presentations of the directly conditioned light *and* when tested with presentations of the preconditioned sound.

## Experiment 3A: Rats exposed to eight sound-light pairings in stage 1

### Conditioning

The baseline levels of freezing during conditioning and test sessions were low (<10%) and did not differ between groups (largest $F_{(1,20)}$ = 2.925; p=0.103). Conditioning was successful. The mean (± SEM) levels of freezing to the light on its final pairing with shock were 48 ± 10.0% in Group 8-IPSI and 58.3 ± 9.7% in Group 8-CONTRA. Freezing to the light increased across the four light-shock pairings in stage 2 ($F_{(1,20)}$ = 28.836; p<0.001; $n_p^2$ = 0.590; 95% CI: [0.742, 1.684]). The rate of this increase did not differ between groups ($F_{(1,20)}$ = 0.469; p=0.501), and there were no significant between-group differences in overall levels of freezing ($F_{(1,20)}$ = 0.480; p=0.496).

### Test

*Figure 4B* shows mean (± SEM) levels of freezing averaged across the eight test presentations of the sound alone (left panel) and light alone (right panel) in Experiment 3A. Both groups showed equivalent levels of freezing to the light ($F_{(1,20)}$ = 2.819; p=0.109). However, rats that received an infusion of muscimol into the contralateral BLA (Group 8-CONTRA) showed significantly lower levels of freezing to the preconditioned sound than rats that received an infusion of muscimol into the ipsilateral BLA (Group 8-IPSI; $F_{(1,20)}$ = 12.883; p=0.002; $n_p^2$ = 0.392; 95% CI: [0.398, 1.501]).

## Experiment 3B: Rats exposed to 32 sound-light pairings in stage 1

### Conditioning

The baseline levels of freezing during conditioning and test sessions were low (<10%) and did not differ between groups ($F_{(1,18)}$ = 0.494; p=0.491). Conditioning was successful. The mean (± SEM) levels of freezing to the light on its final pairing with shock were 42 ± 12.5% in Group 32-IPSI and 50 ± 13.4% in Group 32-CONTRA. Freezing to the light increased across its four pairings with shock in stage 2 ($F_{(1,18)}$ = 22.054; p<0.001; $n_p^2$ = 0.551; 95% CI: [0.640, 1.677]). The rate of this increase did not differ between groups ($F_{(1,18)}$ = 0.957; p=0.341), and there were no significant group differences in overall levels of freezing ($F_{(1,18)}$ = 0.691; p=.417).

### Test

*Figure 4D* shows mean (± SEM) levels of freezing averaged across the eight test presentations of the sound alone (left panel) and light alone (right panel) in Experiment 3B. The statistical analysis confirmed what is evident in the figure: both groups showed equivalent levels of freezing to the directly conditioned light ($F_{(1,18)}$ = 0.002; p=0.965) and to the sensory preconditioned sound ($F_{(1,18)}$ = 0.089; p=0.769).

This experiment has shown that communication between the PRh and BLA is not necessary for the formation of a direct light-shock fear memory: regardless of the number of sound-light pairings in stage 1, disconnecting the PRh and BLA in stage 2 had no effect on freezing to the light across its pairings with shock or when it was presented alone at test. This communication is also not needed for expression of fear to the sound among rats exposed to many sound-light pairings in stage 1: that is, after 32 sound-light pairings, disconnecting the PRh and BLA in stage 2 had no effect on the level

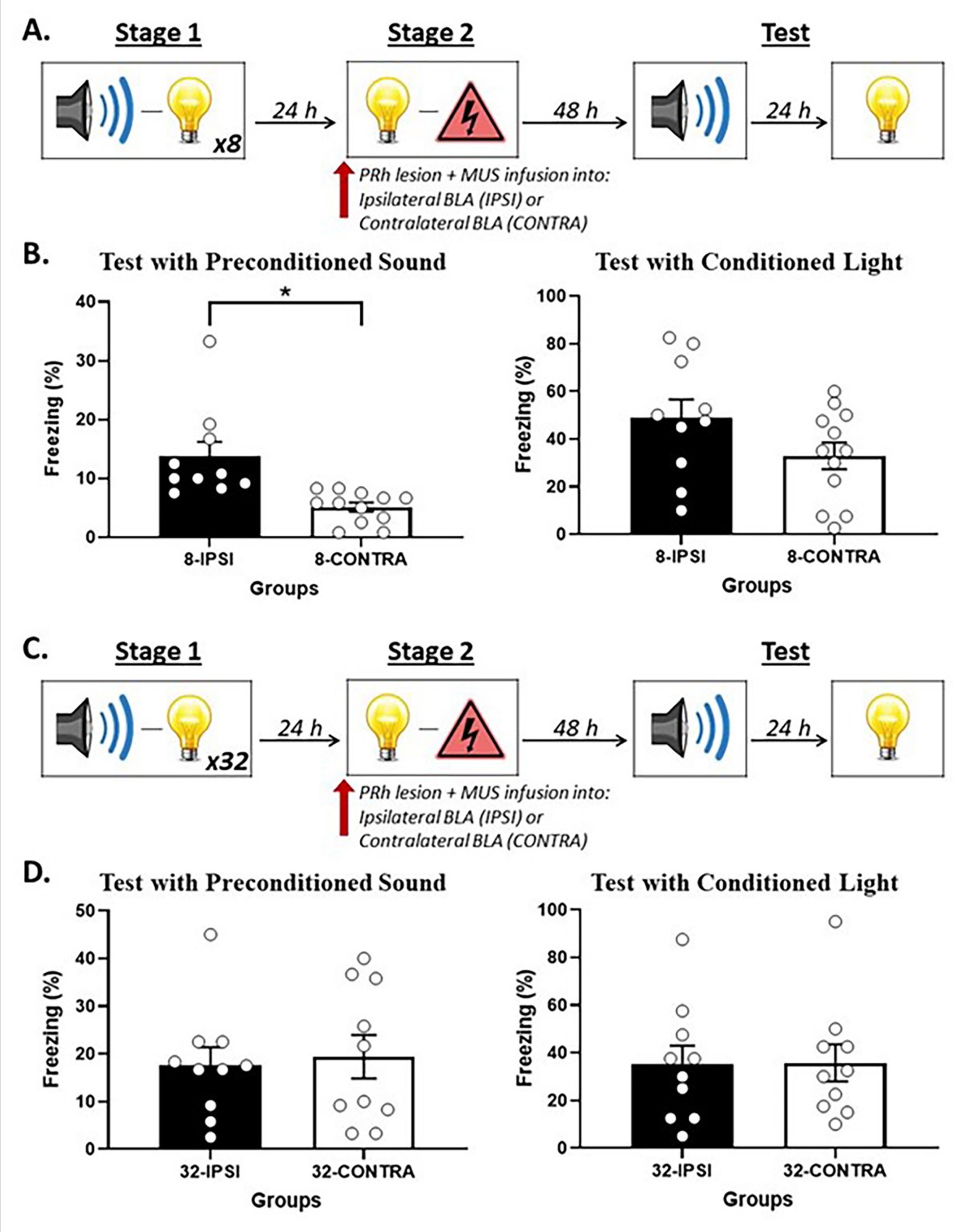

**Figure 4.** Perirhinal cortex-basolateral amygdala complex (PRh-BLA) communication is required for integration in stage 2 among rats that received just a few sound-light pairings in stage 1 but NOT among rats that received many sound-light pairings. (**A**) Schematic of the behavioral procedure for Experiment 3A (Group 8-IPSI, n = 10; and Group 8-CONTRA, n = 12). The red arrow indicates that the infusion of muscimol occurred before stage 2. (**B**) Percentage freezing to the preconditioned sound (left panel) and to the conditioned light (right panel), averaged across the eight trials of their

*Figure 4 continued on next page*

*Figure 4 continued*

respective tests. Data shown are means ± SEM. The asterisk (*) denotes a statistically significant difference (p<0.05). (**C**) Schematic of the behavioral procedure for Experiment 3B (Group 32-IPSI, n = 10; and Group 8-CONTRA, n = 10). The red arrow indicates that the infusion of muscimol occurred before stage 2. (**D**) Percentage freezing to the preconditioned sound (left panel) and to the conditioned light (right panel), averaged across the eight trials of their respective tests. Data shown are means ± SEM.

of freezing to the sound during its testing in stage 3. This communication is, however, needed for expression of fear to the sound among rats exposed to just a few sound-light pairings in stage 1: that is, after eight sound-light pairings, disconnecting the PRh and BLA in stage 2 disrupted freezing to the sound during its testing in stage 3. These results are consistent with our hypotheses that (1) after few sound-light pairings, formation of the mediated sound-shock memory is supported by communication between the PRh and BLA during conditioning of the light; and (2) after many sound-light pairings, chaining of the sound-light and light-shock memories occurs independently of communication between the PRh and BLA during conditioning of the light.

## Experiments 4A and 4B: After many sound-light pairings, integration requires communication between the PRh and BLA at test; after a few sound-light pairings, it does not

We next examined whether communication between the PRh and BLA is necessary for chaining of the sound-light and light-shock memories during testing with the sound. In each of Experiments 4A and 4B, rats underwent surgery in which an excitotoxic lesion of the PRh was created in one hemisphere and a single cannula was implanted in the BLA in either the ipsilateral hemisphere (Groups IPSI) or contralateral hemisphere (Groups CONTRA). After recovery, these rats were exposed to either 8 (Experiment 4A) or 32 (Experiment 4B) sound-light pairings in stage 1, followed a day or so later by a session of light-shock pairings in stage 2. Two days later, all rats were tested with the sound alone and, the next day, with the light alone. Immediately prior to each test session, rats received a BLA infusion of the GABA$_A$ receptor agonist, muscimol. As noted previously, this infusion transiently disconnects the PRh and BLA for rats in Groups CONTRA, as these rats do not have a functioning PRh *and* BLA in either hemisphere, but spares PRh-BLA connectivity for rats in Groups IPSI, as these rats have a functioning PRh and BLA in one hemisphere.

We hypothesized that, among rats exposed to eight sound-light pairings in stage 1 (Experiment 4A), the mediated sound-shock memory that forms in stage 2 may require communication between the PRh and BLA for its retrieval/expression in stage 3. We additionally hypothesized that, among rats exposed to 32 sound-light pairings in stage 1 (Experiment 4B), chaining of the sound-light and light-shock memories at test would also require communication between the PRh and BLA at the time of testing with the sound: for example, because the sound activates the PRh-dependent sound-light memory; which, in turn, activates the BLA-dependent light-shock memory. Hence, we expected that the test level of freezing to the sound would be impaired by the PRh-BLA disconnection in Groups 8-CONTRA (Experiment 4A) and 32-CONTRA (Experiment 4B). That is, relative to rats in Groups 8-IPSI, we expected that rats in Groups 8-CONTRA and 32-CONTRA would freeze just as much when tested with presentations of the directly conditioned light but significantly less when tested with presentations of the preconditioned sound.

## Experiment 4A: Rats exposed to eight sound-light pairings in stage 1
### Conditioning

The baseline levels of freezing during conditioning and test sessions were low (<10%) and did not differ between groups (largest $F_{(1,22)}$ = 3.155; p=0.090). Conditioning was successful. The mean (± SEM) levels of freezing to the light on its final pairing with shock were 66.7 ± 9.0% in Group 8-IPSI and 51.7 ± 11.7% in Group 8-CONTRA. Freezing to the light increased across the four light-shock pairings in stage 2 ($F_{(1,22)}$ = 58.086; p<0.001; $n_p^2$ = 0.725; 95% CI: [1.138, 1.989]). The rate of this increase did not differ between groups ($F_{(1,22)}$ = 2.042; p=0.167), and there were no significant between-group differences in overall levels of freezing ($F_{(1,22)}$ = 1.420; p=0.246).

## Test

*Figure 5B* shows mean (± SEM) levels of freezing averaged across the eight test presentations of the sound alone (left panel) and light alone (right panel) in Experiment 4A. The statistical analysis confirmed what is evident in the figure: both groups showed equivalent levels of freezing to the light ($F_{(1,22)}$ = .131; p=0.721) and to the sound ($F_{(1,22)}$ = 0.186; p=0.670).

## Experiment 4B: Rats exposed to 32 sound-light pairings in stage 1

### Conditioning

The baseline levels of freezing during conditioning and test sessions were low (<10%) and did not differ between groups (largest $F_{(1,18)}$ = 1.956; p=0.176). Conditioning was successful. The mean (± SEM) levels of freezing to the light on its final pairing with shock were 62.5 ± 12.8% in Group 32-IPSI and 60 ± 7.8% in Group 32-CONTRA. Freezing to the light increased across its four pairings with shock in stage 2 ($F_{(1,18)}$ = 76.364; p<0.001; $n_p^2$ = 0.809; 95% CI: [1.724, 2.815]). The rate of this increase did not differ between groups ($F_{(1,18)}$ = 0.041; p=0.842), and there were no significant group differences in overall levels of freezing ($F_{(1,18)}$ = 0.142; p=0.711).

### Test

*Figure 5D* shows mean (± SEM) levels of freezing averaged across the eight test presentations of the sound alone (left panel) and light alone (right panel) in Experiment 4B. Both groups showed equivalent levels of freezing to the directly conditioned light ($F_{(1,18)}$ = 0.033; p=0.858). However, Group 32-CONTRA showed significantly lower levels of freezing to the sensory preconditioned sound compared to Group 32-IPSI ($F_{(1,18)}$ = 7.963; p=0.011; $n_p^2$ = 0.307; 95% CI: [0.178, 1.218]).

This experiment has shown that communication between the PRh and BLA is not necessary for the retrieval/expression of a direct light-shock fear memory: regardless of the number of sound-light pairings in stage 1, disconnecting the PRh and BLA prior to testing had no effect on freezing to the light. This communication is also not needed for expression of fear to the sound among rats exposed to just a few sound-light pairings in stage 1: that is, after eight sound-light pairings, disconnecting the PRh and BLA prior to testing had no effect on the level of freezing to the sound. By contrast, communication between the PRh and BLA is needed for expression of fear to the sound among rats exposed to many sound-light pairings in stage 1: that is, after 32 sound-light pairings, disconnecting the PRh and BLA prior to testing disrupted freezing to the sound. We take these results to mean that communication between the PRh and BLA is *not* needed for retrieval/expression of the mediated sound-shock memory after few sound-light pairings; but *is* needed for chaining of the sound-light and light-shock memories at test after many sound-light pairings. They are considered further in 'Discussion'.

## Discussion

*Wong et al., 2019* provided evidence for online integration of sensory and emotional memories in the rat medial temporal lobe. Specifically, they showed that rats integrate a past sound-light memory with an emotional light-shock memory during formation of the latter; and that this integration takes the form of a mediated sound-shock association. The present study has advanced the Wong et al. findings in three ways. First, it has shown that the way in which rats integrate the sound-light and light-shock memories depends on the number of sound-light pairings in stage 1: after just a few sound-light pairings, integration is achieved through mediated learning about the sound in stage 2; and after many sound-light pairings, integration is achieved through chaining of the sound-light and light-shock memories during testing with the sound. Second, it has shown that mediated conditioning of the sound involves activation of NMDA receptors in the PRh. Mediated conditioning also involves communication between the PRh and BLA across the light-shock pairings but does not require such communication for expression of fear to the sound at test. Third, it has shown that, when integration occurs through chaining of the sound-light and light-shock memories at test, this process also involves communication between the PRh and BLA.

Why does increasing the number of sound-light pairings change the way that rats integrate the sound-light and light-shock memories? One possibility is that increasing the number of sound-light pairings in stage 1 reduces the ability of each stimulus to activate the memory of the other. This is consistent with findings by *Holland, 1998*, who showed that the likelihood of mediated learning in

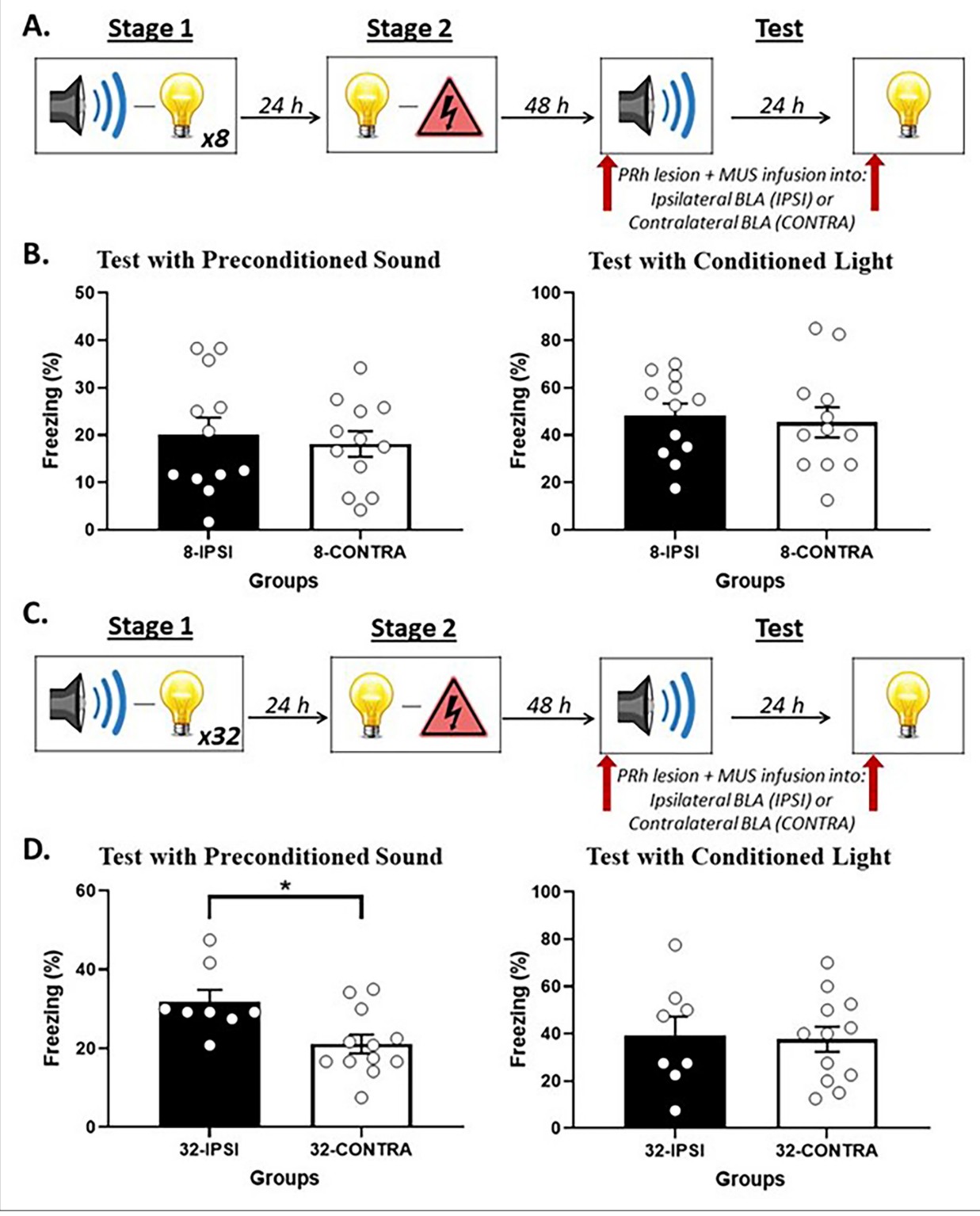

**Figure 5.** Perirhinal cortex-basolateral amygdala complex (PRh-BLA) communication is required for integration at test among rats that received many sound-light pairings in stage 1 but NOT among rats that received few sound-light pairings. (**A**) Schematic of the behavioral procedure for Experiment 4A (Group 8-IPSI, n = 12; Group 8-CONTRA, n = 12). The red arrows indicate that the infusion of muscimol occurred before the test session of the preconditioned sound and before the test session of the conditioned light. (**B**) Percentage freezing to the preconditioned sound (left panel) and to the conditioned light (right panel), averaged across their eight respective test trials. Data shown are means ± SEM. (**C**) Schematic of the behavioral procedure for Experiment 4B (Group 32-IPSI, n = 8; Group 32-CONTRA, n = 12). The red arrows indicate that the infusion of muscimol occurred before

*Figure 5 continued on next page*

*Figure 5 continued*

the test session of the preconditioned sound and before the test session of the conditioned light. (D) Percentage freezing to the preconditioned sound (left panel) and to the conditioned light (right panel), averaged across the eight trials of their respective tests. Data shown are means ± SEM. The asterisks (*) denotes a statistically significant difference (p<0.05).

rats decreases with the amount of training (see also *Holland, 2005*); but inconsistent with our findings that, after extended training, rats continue to integrate the sound-light and light-shock associations through chaining at the time of testing (as chaining is predicated on the sound activating the memory of the light after extended training). Instead, we propose that the change in integration occurs because the increased number of sound-light pairings allows the rats to learn about the order in which the sound and light are presented (*Figure 1*; for evidence that rats acquire order information in sensory preconditioning, see *Barnet et al., 1997*; *Hart et al., 2022*; *Leising et al., 2007*; *Miller and Barnet, 1993*). This order hypothesis is consistent with evidence showing that the way in which animals represent an audio-visual compound changes across repeated compound exposures (e.g., *Bellingham and Gillette, 1981*; *Holmes and Harris, 2009*). It can be tested using a so-called 'backward' sensory preconditioning protocol, which reverses the order of stimulus presentations in stage 1 (e.g., *Ward-Robinson and Hall, 1996*). That is, rather than rats being exposed to the 'forward' sound-light pairings used here and by *Wong et al., 2019*, rats in a backward protocol are exposed to light-sound pairings. Increasing the number of light-sound pairings in this protocol should result in rats learning that the light is followed by the sound (light→sound) and that the sound is followed by nothing (sound→nothing). Hence, during the session of light-shock pairings in stage 2, the light should continue to activate the memory of the sound, resulting in formation of the mediated sound-shock association (e.g., *Ward-Robinson and Hall, 1996*). That is, if our order hypothesis is correct, increasing the number of light-sound pairings in the backward protocol should preserve the likelihood of mediated learning in stage 2 and, if anything, diminish the likelihood of chaining at test in stage 3 (as the sound is never followed by a light). Hence, PRh manipulations that fail to affect fear of the sound when administered after many sound-light pairings (e.g., infusion of DAP5) should disrupt that fear when administered after many light-sound pairings in the backward protocol. This will be assessed in future work.

The present findings advance those of *Wong et al., 2019* by showing that, after just a few sound-light pairings in stage 1, formation of the mediated sound-shock association requires activation of NMDA receptors in the PRh, and communication between the PRh and BLA during conditioning of the light. Communication between the PRh and BLA is likely required so that information about the sound, which is encoded in the PRh during stage 1, can be integrated with information about the shock, which is processed in the BLA. It may also be required to consolidate the mediated sound-shock association in long-term memory. That is, we have previously shown that neuronal activity in the PRh is necessary for consolidation of the mediated sound-shock association in stage 2 as well as retrieval/expression of this association at the time of testing in stage 3: that is, rats do not display fear responses to the sound when activity in the PRh is silenced immediately after the session of light-shock pairings in stage 2 (*Wong et al., 2019*), or prior to testing with the sound alone in stage 3 (*Holmes et al., 2013*; *Wong et al., 2019*). Hence, communication between the PRh and BLA is not only needed for acquisition of the mediated sound-shock association: it must also be needed to consolidate the mediated association through changes in activity in the PRh. Accordingly, we propose that formation of the mediated sound-shock association requires bidirectional communication between the PRh and BLA: acquisition of the mediated sound-shock association requires communication from the PRh-to-BLA, which is the presumed locus of the mediated sound-shock association, whereas consolidation of this association requires communication from the BLA-to-PRh, which ultimately stores the mediated sound-shock association in long-term memory.

While the PRh and BLA clearly communicate to support mediated learning about the sound, this communication is not required for retrieval/expression of the mediated sound-shock association at the time of testing. This result is somewhat surprising as activity in the PRh is needed for expression of fear to the sound (*Holmes et al., 2013*; *Wong et al., 2019*) and raises the question: how does the PRh-dependent sound-shock association come to be expressed in fear responses? One possibility is through projections from the PRh to other sub-territories of the amygdala. For example, in addition to its strong reciprocal connections with the BLA, the PRh projects to the central nucleus of the amygdala

(CeA; *McDonald, 1998*; *Pitkänen et al., 2000*; but see *Shi and Cassell, 1999*), which interacts with midbrain regions (e.g., periaqueductal gray, hypothalamus; *LeDoux et al., 1988*; *Rizvi et al., 1991*; *Weera et al., 2021*) to coordinate different components of the fear response (autonomic, endocrine, behavioral). Importantly, this projection from the PRh-to-CeA is unidirectional and, to the best of our knowledge, has not been assessed for its involvement in sensory preconditioning. We hypothesize that this pathway is the means by which the mediated sound-shock association comes to be expressed in fear responses; and, further, that the PRh-to-BLA and PRh-to-CeA pathways may be differentially involved in the acquisition and retrieval/expression of the mediated sound-shock association. Formation of the mediated sound-shock association in stage 2 of the preconditioning protocol requires communication between the PRh and BLA but may not require communication between the PRh and CeA. By contrast, retrieval/expression of the mediated sound-shock association during testing in stage 3 does not require communication between the PRh and BLA but may require communication between the PRh and CeA.

The present study has also shown that, after many sound-light pairings in stage 1, communication between the PRh and BLA is needed for chaining of the sound-light and light-shock memories at the time of testing with the sound. Importantly, this communication requirement was specific to the time of testing: disconnecting the PRh and BLA across the session of light-shock pairings had no effect on fear of the sound or light at test. It was also specific to the generation of fear responses to the sound: disconnecting the PRh and BLA prior to testing in stage 3 had no effect on fear responses to the directly conditioned light. We propose that, after many sound-light pairings, communication between the PRh and BLA is needed for completion of the sound-light-shock associative chain at test, as the first link in this chain (sound-light) was encoded in the PRh in stage 1 and the second link (light-shock) was encoded in the BLA in stage 2. Future work will examine whether communication between the PRh and BLA is always required for this type of integration (chaining at test). Specifically, we will examine whether communication between the PRh and BLA is required for chaining of the sound-light and light-shock memories under circumstances where the sound-light memory was not encoded in the PRh: for example, after just one sound-light pairing in stage 1 (*Qureshi et al., 2023*), or when the sound-light pairings are administered in a dangerous context (*Holmes et al., 2013*; *Holmes et al., 2018*). We will also examine whether integration always occurs through chaining after many sound-light pairings, or whether there are manipulations that can restore the potential for mediated learning (e.g., manipulations that restore the relative novelty of the sound and light in stage 2, such as the lapse of time).

Finally, it is important to note that the procedure used to disconnect the PRh and BLA permits inferences about whether these regions communicate to support sensory preconditioned fear but leaves open the question of whether any communication requirement is direct or indirect. Indeed, several brain regions have been shown to regulate aspects of sensory preconditioned fear in rats, including the orbitofrontal cortex, prelimbic cortex and dorsal hippocampus (for a review, see *Holmes et al., 2022b*); and the two cortical regions have direct, reciprocal connections with *both* the PRh and BLA. As such, it remains for future work to determine whether the PRh and BLA communicate directly or indirectly to support mediated learning and/or chaining of the sound-light and light-shock memories at test; and, if found to communicate indirectly, whether the indirect communication involves neuronal activity in the orbitofrontal and/or prelimbic cortex.

In summary, the present study has advanced the *Wong et al., 2019* findings by showing that how rats integrate sensory and emotional memories in sensory preconditioning depends on the novelty/familiarity of the auditory and visual stimuli. When the stimuli are relatively novel (i.e., after just a few sound-light pairings in stage 1), the sensory and emotional memories are integrated through mediated learning, and when the stimuli are relatively familiar (i.e., after many sound-light pairings in stage 1), those same memories are integrated through chaining at test. It has also shown that both forms of integration are critically dependent on communication between two regions of the medial temporal lobe, the PRh and BLA; and that this communication is specifically needed for integration of the sensory and emotional memories: that is, communication between the PRh and BLA was not needed for acquisition, consolidation, or retrieval/expression of the emotional light-shock memory. Future work will examine whether this communication requirement is direct or indirect, and how the PRh and BLA work with other regions of the brain (e.g., orbitofrontal cortex, prelimbic cortex, hippocampus) to achieve successful integration.

## Materials and methods

### Subjects

Subjects were experimentally naïve male and female Long-Evans rats, obtained from a colony maintained by the Biological Resources Centre at the University of New South Wales. The rats were housed by sex in plastic cages (40 cm wide × 22 cm high × 67 cm long), with four rats per cage and food and water continuously available. The cages were located in an air-conditioned and humidity-controlled colony room maintained on a 12 h light-dark cycle (lights on at 7 a.m. and off at 7 p.m.) at a temperature of approximately 21°C.

### Surgery

Prior to behavioral training and testing in Experiments 2A and 2B, rats were surgically implanted with cannulas targeting the PRh. Rats were anesthetized using isoflurane, which was delivered in a steady stream of oxygen. The rat was then mounted onto a stereotaxic apparatus (David Kopf Instruments) and incisions made over the skull. Two holes were drilled through the skull and 26-gauge guide cannulas (Plastics One) were implanted into the brain, one in each hemisphere. The tips of the cannulas targeted the PRh at coordinates 4.30 mm posterior to Bregma, 5.00 mm lateral to the midline, 8.4 mm ventral to Bregma, and angled at approximately 9° (*Paxinos and Watson, 2006*). Guide cannulas were secured in place with four jeweller's screws and dental cement. A dummy cannula was kept in each guide cannula at all times except during drug infusions. Immediately after surgery, rats received a subcutaneous (SC) injection of a prophylactic dose (0.1 ml/kg) of Duplocillin (Merck & Co, NJ, USA). Rats were allowed 7 days to recover from surgery, during which they were monitored and weighed daily.

Prior to behavioral training and testing in Experiments 3A, 3B, 4A, and 4B, rats underwent surgery for excitotoxic lesion of the PRh in the right hemisphere and the implantation of a single 26-gauge cannula (Plastics One) targeting the BLA. Excitotoxic lesion of the PRh was achieved via two microinjections of 0.09 M NMDA. These injections were made using a 1 µl Hamilton syringe fixed to an infusion pump (Harvard Apparatus). The pump was programmed to inject a total of 0.25 µl at a rate of 0.1 µl/min. The needle remained in place for an additional 5 min after injection was complete to allow for diffusion of the drug. The tip of the needle targeted the PRh at coordinates 4.00 mm and 4.50 mm posterior to Bregma, 5.00 mm and 5.1 mm lateral to the midline, 8.9 mm ventral to Bregma, and angled at approximately 9° (*Paxinos and Watson, 2006*). The cannula targeted the BLA at coordinates 2.4 mm posterior to Bregma, 4.9 mm lateral to the midline, and 8.4 mm ventral to Bregma (*Paxinos and Watson, 2006*). For rats in the contralateral condition (Groups CONTRA), the cannula was implanted into the opposite hemisphere to the lesion, and for rats in the ipsilateral condition (Groups IPSI), the cannula was implanted into the same hemisphere as the lesion.

### Drug infusions

In Experiments 2A and 2B, the NMDA receptor antagonist, DAP5, or vehicle was infused bilaterally into the PRh. In Experiments 3A, 3B, 4A, and 4B, muscimol or vehicle was infused unilaterally into the BLA. For these infusions, infusion cannulas were connected to 25 µl Hamilton syringes via polyethylene tubing. These syringes were fixed to an infusion pump (Harvard Apparatus). The infusion procedure began by removing the dummy caps from the guide cannulas on each rat and inserting 33-gauge infusion cannulas in its place. The pump was programmed to infuse a total of 0.5 µl of DAP5 at a rate of 0.25 µl/min, and a total of 0.3 µl of muscimol at a rate of 0.1 µl/min, which resulted in a total infusion time of 2 min and 3 min, respectively. The infusion cannulas remained in place for an additional 2 min after the infusion was complete to allow for diffusion of the drug into the PRh or BLA tissue and to avoid reuptake when the infusion cannula was withdrawn. This resulted in a total infusion time of 4 min for DAP5 and 5 min for muscimol. After the additional 2 min, the infusion cannulas were removed and replaced with dummy cannulas. The day prior to infusions, the dummy cannula was removed and the infusion pump was activated to familiarize the rats with the procedure and thereby minimize any effect of procedure on the day of infusions.

### Drugs

NMDA (Sigma, Australia) was dissolved in 10× phosphate-buffered solution to yield a final concentration of 0.09 M and injected into the PRh during surgery. The NMDA receptor antagonist, DAP5

(Sigma) was prepared in the manner described by *Williams-Spooner et al., 2022*. Briefly, it was dissolved in ACSF to a concentration of 10 μg/μl and injected into the PRh as described above. The GABA$_A$ receptor agonist, muscimol (Tocris Bioscience, UK), and prepared in the manner described by *Parkes and Westbrook, 2010*. Briefly, it was dissolved in non-pyrogenic saline (0.9% w/v) to a stock concentration of 5 mg/ml, diluted to a final concentration of 1 mg/ml, and injected into the BLA as described above.

## Histology

After behavioral training and testing, rats were euthanized with a lethal dose of sodium pentobarbital. The brains were extracted and sectioned into 40 μm coronal slices. Every second slice was mounted onto a glass microscope slide and stained with cresyl violet. The placement of the cannula tip was determined under a microscope using the boundaries defined by *Paxinos and Watson, 2006*. Rats with misplaced cannulas were excluded from statistical analysis. *Figure 6* shows placement of the most ventral portion of these cannulas in the PRh for all rats that were included in the statistical analyses of data from Experiments 2A and 2B. *Figures 7 and 8* shows the approximate lesion size in the PRh and the most ventral point of the cannulas in the BLA, respectively, for all rats included in the statistical analysis from Experiments 3A, 3B, 4A, and 4B. The numbers of rats excluded from each experiment based on misplaced cannulas, lesions, and/or health issues were 8 rats in Experiment 2A, 12 rats in Experiment 2B, 10 rats in Experiment 3A, 5 rats in Experiment 3B, 15 rats in Experiment 4A, and 8 rats in Experiment 4B. The final *n*s for each group are shown in the figure legend for each experiment.

## Behavioral apparatus

All experiments were conducted in four identical chambers (30 cm wide × 26 cm long × 30 cm high). The side walls and ceiling were made of aluminum, and the front and back walls of clear plastic. The floor consisted of stainless steel rods, each two mm in diameter and spaced 13 mm apart (center-to-center). A waste tray containing bedding material was located under the floor. At the end of each session, the chambers were cleaned with water and any soiled bedding was removed from the waste trays and replaced with fresh bedding. Each chamber was enclosed in a sound- and light-attenuating wooden cabinet. The cabinet walls were painted black. A speaker and LED lights within a fluorescent tube were mounted onto the back wall of each cabinet. The speaker was used to deliver a 1000 Hz square-wave tone stimulus, presented at 75 dB when measured at the center of the chamber (digital sound meter: Dick Smith Electronics, Australia). The LED lights were used to deliver a flashing light stimulus, presented at 3.5 Hz. A custom-built, constant current generator was used to deliver a 0.5 s duration, 0.8 mA intensity shock to the floor of each chamber. Each chamber was illuminated with an infrared light source, and a camera mounted on the back wall of each cabinet was used to record the behavior of each rat. The cameras were connected to a monitor and a high-definition digital image recorder located in an adjacent room. This room also contained the computer that controlled stimulus and shock presentations through an interface and appropriate software (MATLAB, MathWorks, USA).

## Behavioral procedure

### Context exposure

On days 1 and 2, rats received two 20 min exposures to the chambers, one in the morning and the other approximately 3 h later in the afternoon.

### Sensory preconditioning

Rats received either 8 presentations of the sound and 8 of the light in a single session, or 32 presentations of the sound and 32 of the light across four daily sessions. On day 3, all rats received eight presentations of the sound and eight of the light. Each presentation of the sound was 30 s in duration and each presentation of the light was 10 s in duration. The first stimulus presentation occurred 5 min after rats were placed into the chambers. The offset of one stimulus co-occurred with the onset of the other stimulus for groups that received paired presentations of the sound and the light, while these stimuli were presented separately for groups that received explicitly unpaired presentations. The interval between each paired presentation was 5 min while the interval between each separately presented stimulus was 150 s. After the last stimulus presentation, rats remained in the chambers for

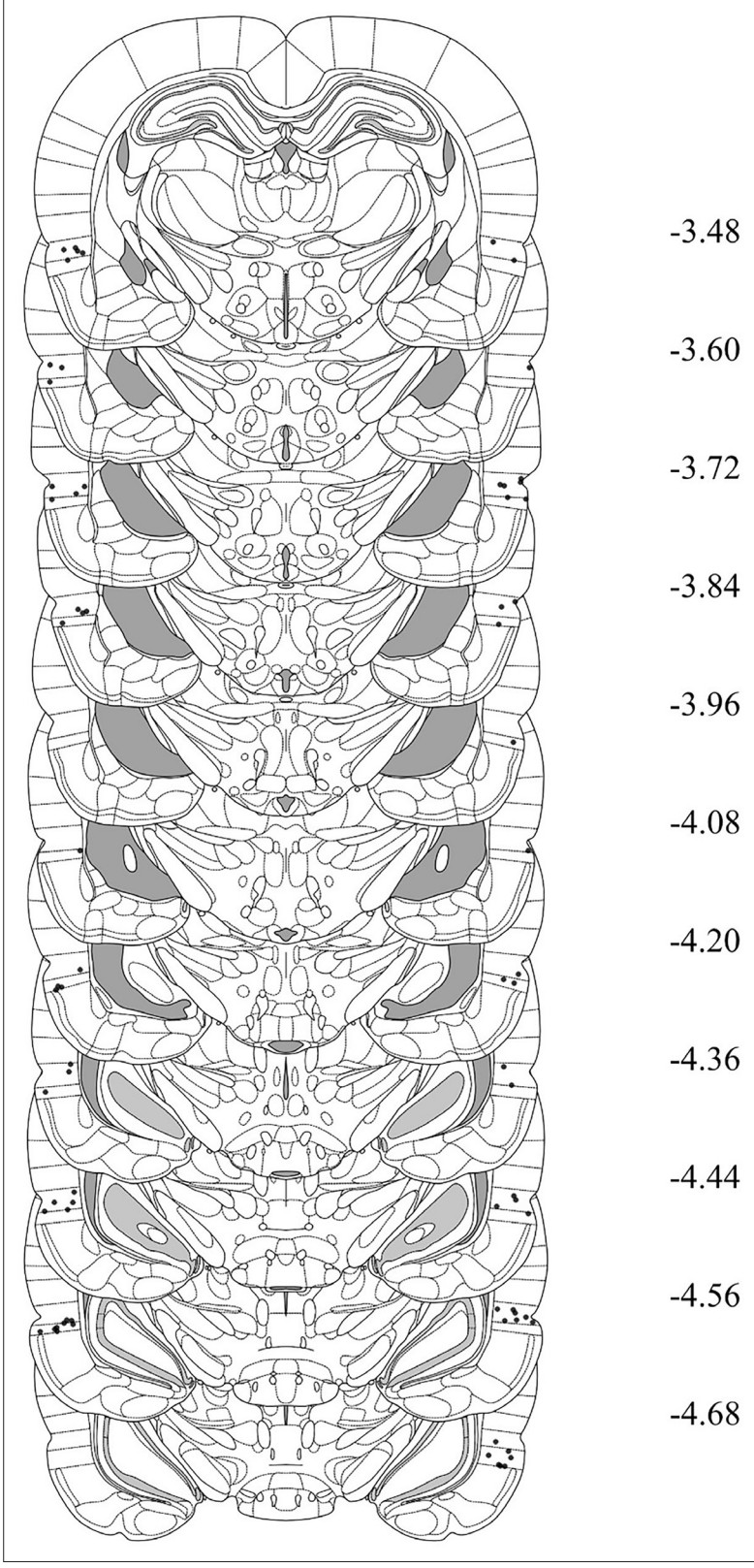

**Figure 6.** Cannula placements in the perirhinal cortex (PRh) taken from rats in Experiments 2A and 2B. The most ventral portion of the cannulas are marked on coronal sections based on the atlas of *Paxinos and Watson, 2006*.

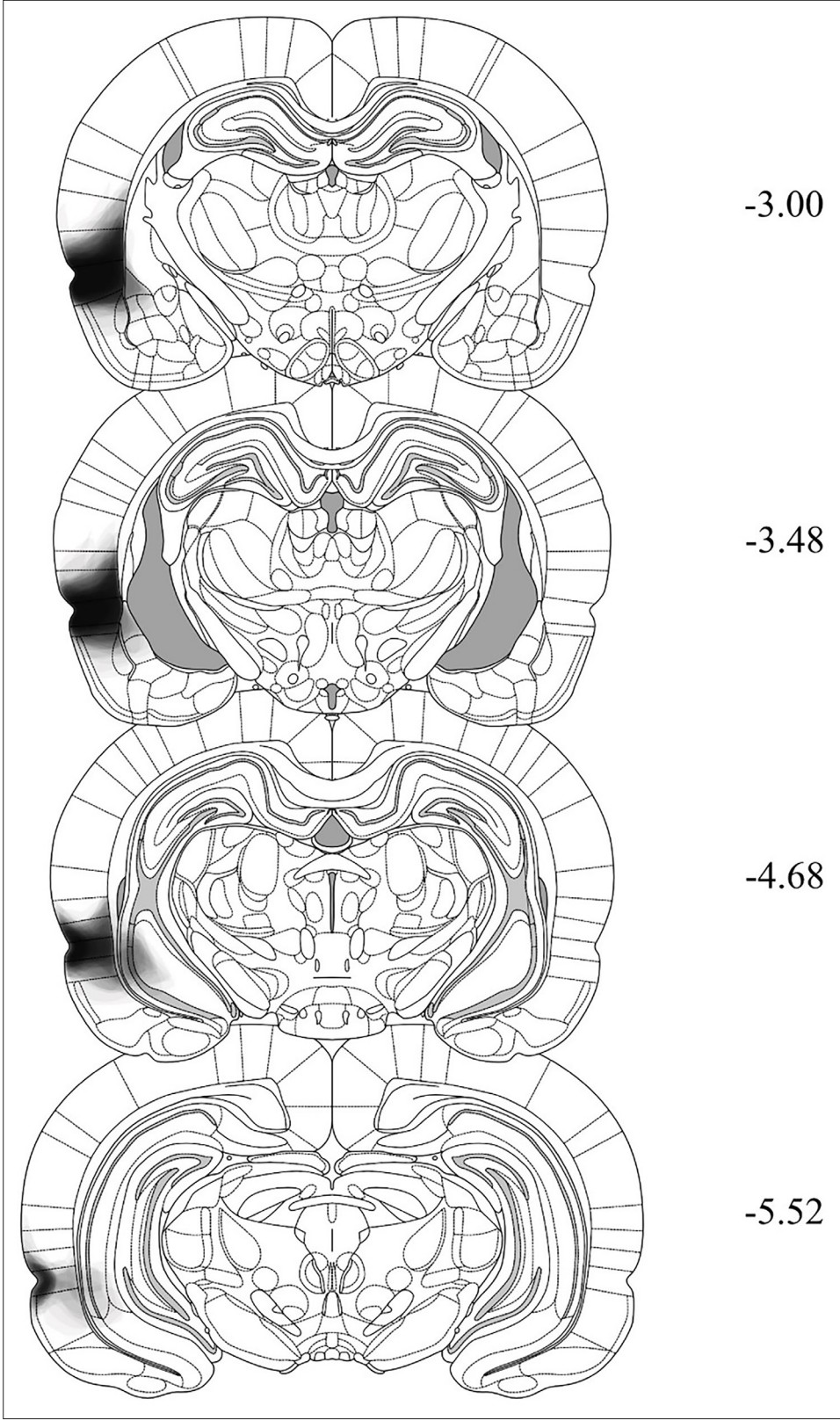

-3.00

-3.48

-4.68

-5.52

**Figure 7.** Histological reconstructions demonstrating the extent of *N*-methyl-d-aspartate (NMDA) lesions in the perirhinal cortex (PRh) taken from rats in Experiments 3A, 3B, 4A, and 4B. The extent is shown across four coronal sections based on the atlas of ***Paxinos and Watson, 2006***. The degree of shading indicates the number of cases exhibiting damage to those regions (e.g., darker areas result from a greater number of cases with damage to that region).

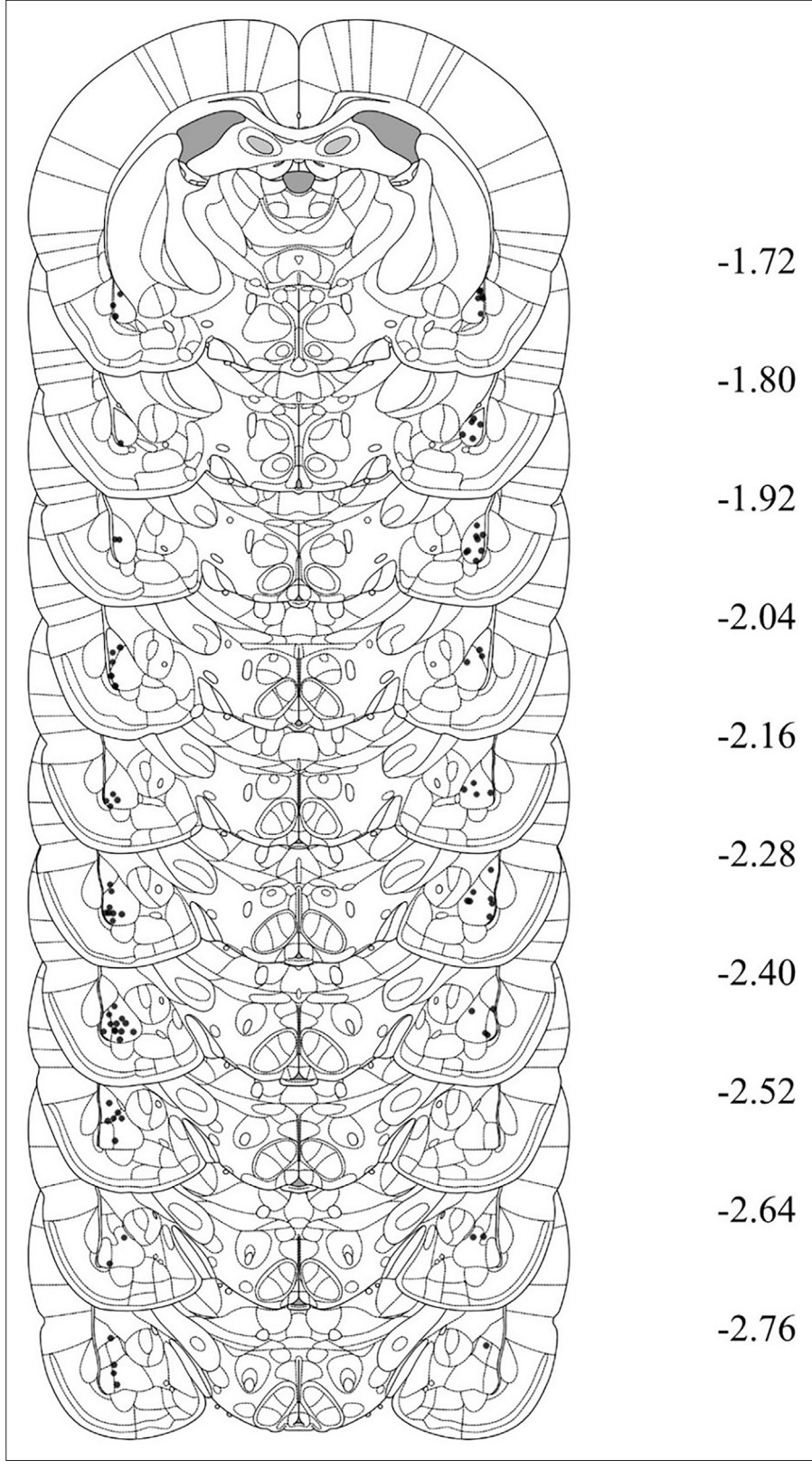

**Figure 8.** Cannula placements in the basolateral amygdala complex (BLA) taken from rats in Experiments 3A, 3B, 4A and 4B. The most ventral portion of the cannulas are marked on coronal sections based on the atlas of *Paxinos and Watson, 2006*.

an additional 1 min. They were then returned to their home cages. This training was repeated on days 4–6 for rats that received 32 presentations of the sound and 32 of the light. All rats proceeded to first-order conditioning (details below) the day after their final session of sound and light exposures, which was day 4 for rats exposed to 8 presentations of the sound and light and day 7 for rats exposed to 32 presentations of the sound and light.

It should be noted that the sound and the light were fully counterbalanced for all experiments. That is, for half of the rats in each group, the sound was the preconditioned stimulus and the light was the conditioned stimulus; while for the remainder, the light was the preconditioned stimulus and the sound was the conditioned stimulus. However, for convenience of explanation, all designs will be described with reference to one half of the counterbalancing: that for which the sound was the preconditioned stimulus and the light was the conditioned stimulus.

### First-order conditioning

On day 4 (8 sound and light exposures) or day 7 (32 sound and light exposures), all rats received four presentations of the light and four presentations of foot shock. The first light presentation occurred 5 min after rats were placed in the chambers. Each 10 s light co-terminated with foot shock and the interval between light-shock pairings was 5 min. Rats remained in the chambers for an additional 1 min after the final stimulus presentation and were then returned to their home cages.

### Context extinction

On day 5 (8 sound and light exposures) or day 8 (32 sound and light exposures), rats received two 20 min exposures to the chambers, one in the morning and the other in the afternoon. These exposures were intended to extinguish any freezing elicited by the context and thereby provide a measure of the level of freezing to the sound and the light unconfounded by context elicited freezing. On day 6 (8 sound and light exposures) or day 9 (32 sound and light exposures), rats received a further 10 min extinction exposure to the context.

### Test

On day 6 (8 sound and light exposures) or day 9 (32 sound and light exposures), approximately 2 h after the context extinction session, rats were tested for their levels of freezing to the preconditioned sound. On day 7 (8 sound and light exposures) or day 10 (32 sound and light exposures), rats were tested for their levels of freezing to the conditioned light. For each test, the first stimulus was presented 2 min after rats were placed in the chambers. Each test consisted of eight stimulus alone presentations, with a 3 min interval between each presentation. Each sound presentation was 30 s in duration and each light presentation was 10 s. Rats remained in the chamber for an additional minute after the final stimulus presentation.

## Scoring and statistics

Conditioning and test sessions were recorded. Freezing, defined as the absence of all movements except those required for breathing (*Fanselow, 1980*), was used as a measure of conditioned fear. Freezing data were collected using a time-sampling procedure in which each rat was scored as either 'freezing' or 'not freezing' every 2 s by an observer blind to the rat's group allocation. A percentage score was then calculated by dividing the number of samples scored as freezing by the total number of samples. The baseline level of freezing was established by scoring the first 2 min at the start of each experimental session: that is, we divided the total number of samples scored as freezing by the total number of observed samples, which was 60. The levels of freezing to the 10 s conditioned stimulus and 30 s preconditioned stimulus were established in a similar manner: we scored the entire period of each stimulus presentation and divided the number of samples scored as freezing by the total number of observed samples, which was 5 for each presentation of the conditioned stimulus and 15 for each presentation of the preconditioned stimulus. A second blind observer scored a randomly selected 50% of the data. The correlation between the scores was high (Pearson > 0.9). The data were analyzed using a set of planned orthogonal contrasts (*Hays, 1967*), with the type 1 error rate controlled at $\alpha = 0.05$. Standardized 95% CIs are reported for significant results and partial eta-squared ($n_p{}^2$) is reported as a measure of effect size (where 0.01, 0.06, and 0.14 is a small, medium, and large effect size, respectively). The required number of rats per group was determined during the design stage of the study.

It was based on our prior studies of sensory preconditioning which indicated that eight subjects per group (or per comparison in the contrast testing procedure) provides sufficient statistical power to detect effect sizes >0.5 with the recommended probability of 0.8–0.9.

## Additional information

### Competing interests

Nathan M Holmes: Reviewing editor, eLife. The other authors declare that no competing interests exist.

### Funding

| Funder | Grant reference number | Author |
|---|---|---|
| Australian Research Council | DP190100747 | Simon Killcross Vincent Laurent R Fred Westbrook Nathan M Holmes |
| Australian Research Council | FT190100697 | Nathan M Holmes |

The funders had no role in study design, data collection and interpretation, or the decision to submit the work for publication.

### Author contributions

Francesca S Wong, Conceptualization, Formal analysis, Investigation, Writing – original draft, Project administration; Alina B Thomas, Formal analysis, Investigation, Methodology; Simon Killcross, Vincent Laurent, Conceptualization, Resources, Funding acquisition, Writing – review and editing; R Fred Westbrook, Nathan M Holmes, Conceptualization, Resources, Funding acquisition, Project administration, Writing – review and editing

### Author ORCIDs

Francesca S Wong https://orcid.org/0000-0002-8533-9833
Vincent Laurent https://orcid.org/0000-0003-2333-8437
Nathan M Holmes https://orcid.org/0000-0002-0592-2026

### Ethics

This study was performed in strict accordance with the recommendations in the Guide for the Care and Use of Laboratory Animals of the National Health and Medical Research Council in Australia. All of the animals were handled according to approved Animal Care and Ethics Committee (ACEC) protocols of the University of New South Wales. The experimental protocols were approved by the UNSW ACEC (Permit Number: 22/84B). All surgery was performed under isoflurane anesthesia, and every effort was made to minimize suffering.

Reviewer #1 (Public review): https://doi.org/10.7554/eLife.101965.3.sa1
Reviewer #2 (Public review): https://doi.org/10.7554/eLife.101965.3.sa2
Reviewer #3 (Public review): https://doi.org/10.7554/eLife.101965.3.sa3
Author response https://doi.org/10.7554/eLife.101965.3.sa4

## Additional files

### Supplementary files

MDAR checklist

## Data availability

All data generated or analysed during this study are included in the manuscript and supporting files; source data files have been provided for Figures 2, 3, 4 and 5 via the Open Science Framework repository (https://osf.io/dfg7x/). These files contain the numerical data used to generate the figures.

The following dataset was generated:

| Author(s) | Year | Dataset title | Dataset URL | Database and Identifier |
|---|---|---|---|---|
| Wong FS, Thomas AB, Killcross AS, Laurent V, Westbrook RF, Holmes NM | 2025 | Integration of sensory and fear memories in the rat medial temporal lobe: advancing research by Wong et al. (2019) | https://osf.io/dfg7x/ | Open Science Framework, dfg7x |

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
