## [Editor Report · eLife Assessment]

This **important** study by Wong et al. addresses a long-standing question in the field of associative learning regarding how a motivationally relevant event can be inferred from prior learning based on neutral stimulus-stimulus associations. The research provides **convincing** behavioral and neurophysiological evidence to address this question. The article will be interesting for researchers in behavioral and cognitive neuroscience.

---

## [Referee Report · Reviewer #1 (Public review)]

Summary:

This study is an important follow-up to their prior work - Wong et al. (2019), starting with clear questions and hypotheses, followed by a series of thoughtful and organized experiments. The method and results are convincing. Experiment 1 demonstrated the sensory preconditioned fear with few (8) or many (32) sound-light pairings. Experiments 2A and 2B showed the role of PRh NMDA receptors during conditioning for online integration, revealing that this contribution is present only after few sound-light pairings, not after many sound-light pairings. Experiments 3A and 3B showed the contribution of PRh-BLA communication to online integration, again only after few but not after many. Contrary to Experiments 3A and 3B, Experiments 4A and 4B showed the contribution of PRh-BLA communication to integration at test only after many but not few sound-light pairings.

Strengths:

Throughout the manuscript, the methods and results are clearly organized and described, and the use of statistics is solid, all contributing to the overall clarity of the research. The discussion section was also well written, effectively comparing the current research with the prior work and offering insightful interpretations and potential future directions for this line of research.

All my previous concerns have been well addressed in this revised version. I do not have further concerns about the current version of this manuscript.

---

## [Referee Report · Reviewer #2 (Public review)]

This manuscript builds on the authors' earlier work, most recently Wong et al. 2019, in which they showed the importance of the perirhinal cortex (PRh) during the first-order conditioning stage of sensory preconditioning. Sensory preconditioning requires learning between two neutral stimuli (S2-S1) and subsequent development of a conditioned response to one of the neutral stimuli after pairing of the other stimulus with a motivationally relevant unconditioned stimulus (S1-US). One highly debated question regarding the mechanisms of learning of sensory preconditioning has been whether conditioned responses evoked by the indirectly trained stimulus (S2) occur through a mediated representation at the time of the first-order US training, or whether the conditioned responses develop through a chained evoked representation (S2 S1  US) at the time of test. The authors' prior findings provided strong evidence for PRh being involved in mediated learning during the first-order training. They showed that protein synthesis was required during the first-order S1-US learning to support the conditioned response to the indirectly trained stimulus (S2) at test.

One question remaining following the previous paper was whether certain conditions may promote a chaining mechanism over mediated learning, as there is some evidence for chained representations at the time of test. In this paper, the authors directly address this important question and find unambiguous results that the extent of training during the preconditioning stage impacts the involvement of PRh during the first-order conditioning or stage 2. They show that putative blockade of synaptic changes in PRh, using an NMDA antagonist, disrupts responding to the preconditioned cue at test during shorter duration preconditioning training (8 trials), but not during extended training (32 trials). They also show that this is the case for communication between the PRh and BLA during the same stage of training using a contralateral inactivation approach. This confirms their previous findings in 2019 of connectivity between these regions for the short duration training, while they observe here for the first time that this is not the case for extended training. Finally, they show that with extended training, communication between BLA and the PRh is required at the final test of the preconditioned stimulus, but not for the short duration training.

Strengths:

The results are clear and extremely consistent across experiments within this paper as well as with earlier work. The experiments here are thorough, well-conceived, and address an important and highly debated question in the field regarding the neural and psychological mechanisms underlying sensory preconditioning. This work is highly impactful for the field as the debate over mediated versus chaining mechanisms has been an important topic for more than 70 years.

Comments on revisions:

Thank you for addressing all of my concerns in considerable detail. I have no more suggestions for the authors. This is a fantastic paper both in the experimental design and the execution as well as in the high quality of writing.

---

## [Referee Report · Reviewer #3 (Public review)]

The authors tested whether: 1. The number of stimulus-stimulus pairings alters whether preconditioned fear depends on online integration during formation of the stimulus-outcome memory or during the probe test/mobilization phase, when the original stimulus, which was never paired with aversive events, elicits fear via chaining of stimulus-stimulus and stimulus-outcome memories. They found that sensory preconditioning was successful with either 8 or 32 stimulus-stimulus pairings. Perirhinal cortex NMDA receptor blockade during stimulus-outcome learning impaired preconditioning following 8 but not 32 pairings during preconditioning. Therefore, perirhinal cortex NMDA activity is required for online integration or mediated learning. Perirhinal-basolateral amygdala had nearly identical effects with the same interpretation: these areas communicate during stimulus-outcome learning, and this online communication is required for later expressing preconditioned fear. Disconnection prior to the probe test, when chaining might occur, had different effects: it impaired the expression of preconditioned fear in rats that received 32, but not 8, pairings during preconditioning. The study has several strengths and provides a thoughtful discussion of future experiments. The study is highly impactful and significant; the authors were successful in describing the behavioral and neurobiological mechanisms of mediated learning versus chaining in sensory preconditioning, which is often debated in the learning field. Therefore this study will have a significant impact on the behavioral neurobiology and learning fields.

Strengths:

Careful, rigorous experimental design and statistics

The discussion leaves open questions that are very much worth exploring. For example - why did perirhinal-amygdala disconnection prior to the probe have no effect in the 8-pairing group, when bilateral perirhinal inactivation did (in Wong et al, 2019)? The authors propose that perirhinal cortex outputs bypass the amygdala during the probe test, which is an excellent hypothesis to test.

The experiments are very explicitly hypothesis-driven, and the authors provide evidence of how and why mediated learning and chaining occur during sensory-sensory learning.

---

## [Author Response]

The following is the authors’ response to the original reviews.

**Public Reviews:**

**Reviewer #1 (Public review):**
Summary:This study is an important follow-up to their prior work - Wong et al. (2019), starting with clear questions and hypotheses, followed by a series of thoughtful and organized experiments. The method and results are convincing. Experiment 1 demonstrated the sensory preconditioned fear with few (8) or many (32) sound-light pairings. Experiments 2A and 2B showed the role of PRh NMDA receptors during conditioning for online integration, revealing that this contribution is present only after a few sound-light pairings, not after many sound-light pairings. Experiments 3A and 3B showed the contribution of PRh-BLA communication to online integration, again only after a few but not after many. Contrary to Experiments 3A and 3B, Experiments 4A and 4B showed the contribution of PRh-BLA communication to integration at test only after many but not few sound-light pairings.Strengths:Throughout the manuscript, the methods and results are clearly organized and described, and the use of statistics is solid, all contributing to the overall clarity of the research. The discussion section was also well-written, effectively comparing the current research with the prior work and offering insightful interpretations and potential future directions for this line of research. I have only a limited amount of concerns about some results and some details of experiments/statistics.

We thank the reviewer for their positive assessment.

Weaknesses:Could you provide further interpretation regarding line 171: the observation that sensory preconditioned fear increased with the number of sound-light pairings? Was this increase due to better sound-light association learning during Stage 1? Additionally, were there any experimental differences between Experiment 1 and the other experiments that might explain why freezing was higher in the P32 group compared to the P8 group? This pattern seemed to be absent in the other experiments. If we consider the hypothesis that the online integration mechanism is more active with fewer pairings and the chaining mechanism at the test is more prominent with many pairings, we wouldn't expect a difference between the P8 and P32 groups. Given the relatively small sample size in Experiment 1, the authors might consider conducting a cross-experiment analysis or something similar to investigate this further.

We appreciate the reviewer’s point and thank them for the question. The heightened level of sensory preconditioned fear among rats that received many sound-light pairings in the initial control experiment (Group P32) may reflect the combined effects of both mediated learning and chaining at test. We are, however, reluctant to offer a strong interpretation of this result as it was not replicated in the subsequent experiments: i.e., the levels of freezing to the sensory preconditioned stimulus at test were almost identical among vehicle-injected controls that received either few (8) or many (32) sound-light pairings in Experiments 2A and 2B; and this was also true in Experiments 3A and 3B, and again in Experiments 4A and 4B. A key difference between the initial and subsequent experiments is that, in contrast to the initial experiment, rats in subsequent experiments underwent surgery for one reason or another (implantation of cannulas, lesion of the perirhinal cortex). The implication is that surgical interventions in the perirhinal cortex and/or basolateral amygdala might affect the way that rats integrate the sound-light and light-shock associations in sensory preconditioning: i.e., they may force rats to rely on one type of integration strategy or the other. This is, of course, purely speculative – it will be addressed in future research.

**Reviewer #2 (Public review):**
This manuscript builds on the authors' earlier work, most recently Wong et al. 2019, in which they showed the importance of the perirhinal cortex (PRh) during the first-order conditioning stage of sensory preconditioning. Sensory preconditioning requires learning between two neutral stimuli (S2-S1) and subsequent development of a conditioned response to one of the neutral stimuli after pairing of the other stimulus with a motivationally relevant unconditioned stimulus (S1-US). One highly debated question regarding the mechanisms of learning of sensory preconditioning has been whether conditioned responses evoked by the indirectly trained stimulus (S2) occur through a mediated representation at the time of the first-order US training, or whether the conditioned responses develop through a chained evoked representation (S2 S1  US) at the time of test. The authors' prior findings provided strong evidence for PRh being involved in mediated learning during the first-order training. They showed that protein synthesis was required during the first-order S1-US learning to support the conditioned response to the indirectly trained stimulus (S2) at the test.One question remaining following the previous paper was whether certain conditions may promote a chaining mechanism over mediated learning, as there is some evidence for chained representations at the time of the test. In this paper, the authors directly address this important question and find unambiguous results that the extent of training during the preconditioning stage impacts the involvement of PRh during the first-order conditioning or stage 2. They show that putative blockade of synaptic changes in PRh, using an NMDA antagonist, disrupts responding to the preconditioned cue at test during shorter duration preconditioning training (8 trials), but not during extended training (32 trials). They also show that this is the case for communication between the PRh and BLA during the same stage of training using a contralateral inactivation approach. This confirms their previous findings in 2019 of connectivity between these regions for the short-duration training, while they observe here for the first time that this is not the case for extended training. Finally, they show that with extended training, communication between BLA and the PRh is required at the final test of the preconditioned stimulus, but not for the short duration training.The results are clear and extremely consistent across experiments within this paper as well as with earlier work. The experiments here are thorough, and well-conceived, and address an important and highly debated question in the field regarding the neural and psychological mechanisms underlying sensory preconditioning. This work is highly impactful for the field as the debate over mediated versus chaining mechanisms has been an important topic for more than 70 years.

We thank the reviewer for their kind assessment.

**Reviewer #3 (Public review):**
The authors tested whether the number of stimulus-stimulus pairings alters whether preconditioned fear depends on online integration during the formation of the stimulus-outcome memory or during the probe test/mobilization phase, when the original stimulus, which was never paired with aversive events, elicits fear via chaining of stimulus-stimulus and stimulus-outcome memories. They found that sensory preconditioning was successful with either 8 or 32 stimulus-stimulus pairings. Perirhinal cortex NMDA receptor blockade during stimulus-outcome learning impaired preconditioning following 8 but not 32 pairings during preconditioning. Therefore, perirhinal cortex NMDA activity is required for online integration or mediated learning. Perirhinal-basolateral amygdala had nearly identical effects with the same interpretation: these areas communicate during stimulus-outcome learning, and this online communication is required for later expressing preconditioned fear. Disconnection prior to the probe test, when chaining might occur, had different effects: it impaired the expression of preconditioned fear in rats that received 32, but not 8, pairings during preconditioning. The study has several strengths and provides a thoughtful discussion of future experiments. The study is highly impactful and significant; the authors were successful in describing the behavioral and neurobiological mechanisms of mediated learning versus chaining in sensory preconditioning, which is often debated in the learning field. Therefore this study will have a significant impact on the behavioral neurobiology and learning fields.Strengths:Careful, rigorous experimental design and statistics.The discussion leaves open questions that are very much worth exploring. For example - why did perirhinal-amygdala disconnection prior to the probe have no effect in the 8-pairing group, when bilateral perirhinal inactivation did (in Wong et al, 2019)? The authors propose that perirhinal cortex outputs bypass the amygdala during the probe test, which is an excellent hypothesis to test.The authors provide evidence that both mediated learning and chaining occur.

Thank you for the positive assessment – we fully intend to identify the circuitry that regulates retrieval/expression of sensory preconditioned fear when it is based on mediated learning in stage 2.

Weaknesses:This is inherent to all neural interference and behavioral experiments: biological/psychological functions do not typically operate binarily. There is no single clear number or parameter at which mediated learning or chaining happens, and both probably happen to some extent. Addressing this is even more difficult given behavioral variability across subjects, implant sites, etc. Thus, this is not so much a weakness particular to this study as much as an existential problem, which the authors were able to work around with careful experimental design and appropriate controls.

We completely agree with the point raised here and thank the reviewer for their assessment.

**Recommendations for the authors:**

**Reviewer #1 (Recommendations for the authors):**
(1) It appears that the method description for Sensory Preconditioning was copied from their previous Wong et al. (2019) paper, which is fine, but in the current research, the authors use 8 or 32 presentations, which is not reflected in the description.

Thank you for bringing this to our attention. This is now addressed in the method section on page 27 (beginning at line 655):

“Rats received either eight presentations of the sound and eight of the light in a single session, or 32 presentations of the sound and 32 of the light across four daily sessions. On Day 3, all rats received eight presentations of the sound and eight of the light. Each presentation of the sound was 30 s in duration and each presentation of the light was 10 s in duration. The first stimulus presentation occurred five min after rats were placed into the chambers. The offset of one stimulus co-occurred with the onset of the other stimulus for groups that received paired presentations of the sound and the light, while these stimuli were presented separately for groups that received explicitly unpaired presentations. The interval between each paired presentation was five min while the interval between each separately presented stimulus was 150 s. After the last stimulus presentation, rats remained in the chambers for an additional one min. They were then returned to their home cages. This training was repeated on Days 4-6 for rats that received 32 presentations of the sound and 32 of the light. All rats proceeded to first-order conditioning (details below) the day after their final session of sound and light exposures, which was Day 4 for rats exposed to eight presentations of the sound and light and Day 7 for rats exposed to 32 presentations of the sound and light.”

(2) Line 148: Could the authors clarify how the "significant linear increase" was assessed? From similar descriptions in later experiments, it seems it was based on a comparison of freezing across the four presentations, but the F(1,26) statistic suggests there seemed to be a half-split test. The same questions exist in all the experiments. Please clarify.

Conditioning data were analysed using contrasts with repeated measures in ANOVA. The repeated measures (or within-subject) factor was “trial” as all rats were exposed to four light-shock pairings in this stage of training. We examined whether there was a significant linear increase in freezing across trials using a standard within-subject contrast. The specific coefficients for this contrast, given the four trials, were -3, -1, 1, and 3. The reason that the degrees of freedom remain 1 and 26 in this analysis is because the within-subject contrast is part of a set of planned orthogonal contrasts. That is, in any planned analysis of the sort conducted here, the df1 will always be 1, indicating the very nature of the analysis. There was no splitting of the data, or comparisons between the split halves.

(3) Line 154: Could the authors clarify what is meant by "other main effects and their interactions"? It is not clearly inferable from the context.

Apologies for the confusion here. “Other main effects” refer to the two between-subject factors in isolation: i.e., the overall comparison of freezing to the light (averaged across the four trials) between groups that received either paired or unpaired stimulus presentations in stage 1 (factor 1 - main effect 1), and between groups that received either eight or 32 sound and light exposures in stage 1 (factor 2 - main effect 2). “Their interaction” refers to the assessment of whether the overall difference in freezing to the light (averaged across the four trials) between Groups P8 and U8 differs from the overall difference in freezing to the light (averaged across the four trials) between Groups P32 and U32. We have edited the text near line 153 to indicate that:

“The overall comparisons of freezing to the light (averaged across the four conditioning trials) between groups that received either paired or unpaired stimulus presentations in stage 1 (factor 1), and between groups that received either eight or 32 sound and light exposures in stage 1 (factor 2), were not significant (Fs < .45, p > .508). The interaction between these two between-subject factors was also not significant (F < .45, p > .508).”

(4) The use of sound and light as preconditioned and conditioned cues are counterbalanced. Was there any difference in the increase of freezing during conditioning depending on the type of conditioned cues? Was there any difference in the preconditioned fear? While it is hard to assess statistical significance due to the sample size limit, even observing a trend could be interesting.

We examined whether the levels of freezing to the conditioned and preconditioned stimuli depend on their physical identity. In general, there was a slight trend towards more freezing to the preconditioned stimulus when it was a tone, and *less* freezing to the conditioned stimulus when it was a tone. These are, however, simply indications. None of the statistical comparisons between rats for which the preconditioned stimulus was the tone (and, thereby, conditioned stimulus was the light) and rats for which the preconditioned stimulus was the light (and, thereby, conditioned stimulus was the tone) reached the conventional level of significance.

(5) General suggestion on reporting non-significant statistics: the authors reported a small F statistic value a few times to suggest non-significance. But without clearly specifying degrees of freedom, it is hard to get a sense of statistical significance (e.g. Line 227, largest F<3.10). I recommend adding p values alongside the F statistics and reporting exact statistics whenever possible.

Apologies for the omission. The p values have now been included alongside all non-significant F statistics.

(6) Another general suggestion is to use non-parametric statistical testing with such small sample sizes. I recommend using the Kruskal-Wallis H test (the non-parametric equivalent of F-statistic) to replace the ANOVA result. Also, given many tests only involve comparing two independent groups, using Mann-Whitney U test (the non-parametric equivalent of independent t-test) would be sufficient.

We understand that small sample sizes can occasionally lead to unequal variances between groups, which necessitates the use of non-parametric statistics. However, as non-parametric statistics raise a different set of issues for data analysis (e.g., power) and interpretation, our general view for the type of data collected in this study is that parametric analyses are appropriate and should be retained (particularly in the absence of unequal variances between groups). We hold this view for two reasons. First, the hypotheses tested in the present series were derived from past work in which parametric analyses revealed meaningful patterns of results at the same level of statistical power. Second, the application of these analyses then yielded results consistent with our hypotheses: for the most part, we observed between-group differences where we expected there to be such differences and did not observe between-group differences where we did not expect there to be such differences. As such, we have not switched from a parametric to non-parametric analysis strategy. We do, however, appreciate the suggestion and will apply a non-parametric approach where it is warranted in our future work.

**Reviewer #2 (Recommendations for the authors):**
I have a few very minor comments for the authors regarding the discussion and interpretation of the very nice experimental results.(1) In Figures 4 and 5, the authors provide a schematic of the experiment. It's very clearly indicated whether the BLA inactivation is ipsi- or contralateral, but the unilateral PRh lesion isn't mentioned. I'd recommend including that here so that someone reading through the figures can more easily understand the experiment. The hypothesis is clear and the experiment is so well designed that a read through of the figures can relay most information to an experienced reader.

Thank you for this suggestion – we have included information about the unilateral PRh lesion in the schematic for Figures 4 and 5.

(2) The authors have an extended description of backward conditioning in the discussion. It seems like the authors are suggesting this as an important future direction, but they never explicitly say this, resulting in a bit of confusion as to what this section refers to. Also, Ward-Robinson and Hall 1996 showed backward sensory preconditioning using a serial auditory-visual association and argued for a mediated solution based on their results. It may be worth citing that paper here.

Apologies for the lack of clarity. We have revised this point in the discussion (page 18, beginning line 434) and referenced Ward-Robinson and Hall (1996):

“Why does increasing the number of sound-light pairings change the way that rats integrate the sound-light and light-shock memories? One possibility is that increasing the number of sound-light pairings in stage 1 reduces the ability of each stimulus to activate the memory of the other. This is consistent with findings by Holland (1998), who showed that the likelihood of mediated learning in rats decreases with the amount of training (see also Holland, 2005); but inconsistent with our findings that, after extended training, rats continue to integrate the sound-light and light-shock associations through chaining at the time of testing (as chaining is predicated on the sound activating the memory of the light after extended training). Instead, we propose that the change in integration occurs because the increased number of sound-light pairings allows the rats to learn about the order in which the sound and light are presented (Figure 1; for evidence that rats acquire order information in sensory preconditioning, see Barnet et al., 1997; Hart et al., 2022; Leising et al., 2007; Miller & Barnet, 1993). This order hypothesis is consistent with evidence showing that the way in which animals represent an audio-visual compound changes across repeated compound exposures (e.g., Bellingham & Gillette, 1981; Holmes & Harris, 2009). It can be tested using a so-called “backward” sensory preconditioning protocol, which reverses the order of stimulus presentations in stage 1 (e.g., Ward-Robinson & Hall, 1996). That is, rather than rats being exposed to the “forward” sound-light pairings used here and by Wong et al. (2019), rats in a backward protocol are exposed to light-sound pairings. Increasing the number of light-sound pairings in this protocol should result in rats learning that the light is followed by the sound (light→sound) and that the sound is followed by nothing (sound→nothing). Hence, during the session of light-shock pairings in stage 2, the light should continue to activate the memory of the sound, resulting in formation of the mediated sound-shock association (e.g., Ward-Robinson & Hall, 1996). That is, if our order hypothesis is correct, increasing the number of light-sound pairings in the backward protocol should preserve the likelihood of mediated learning in stage 2 and, if anything, diminish the likelihood of chaining at test in stage 3 (as the sound is never followed by a light). Hence, PRh manipulations that fail to affect fear of the sound when administered after many sound-light pairings (e.g., infusion of DAP5) should disrupt that fear when administered after many light-sound pairings in the backward protocol. This will be assessed in future work.”

(3) Line 467 in the discussion suggests that the results are surprising that PRh-BLA communication is not needed at test when learning putatively occurs through a mediated mechanism during first-order conditioning. I was a bit surprised by this comment since I was under the assumption that only BLA was required at this point after consolidation of the mediated learning. Holmes et al., 2013 showed that BLA is required for extinction to S2 after first-order conditioning. In that experiment they inactivated BLA during S2- presentations (typically considered the extinction test), and showed that reduction to S2 did not occur the subsequent day, indicating the memory was stored in BLA and may not necessarily require PRh-BLA communication.

The result noted here was somewhat surprising as our past studies showed that silencing activity in the PRh prior to testing attenuates freezing to a sensory preconditioned stimulus (i.e., an S2). We took this to mean that the PRh is necessary for retrieval/expression of fear to S2 and supposed that this retrieval/expression would be achieved through communication between the PRh and BLA. However, the results of the PRh-BLA disconnection at test show that this communication is not required, leaving us to speculate that retrieval/expression of fear to S2 may be achieved through communication between the PRh and CeA.

We have edited the opening of the relevant paragraph to clarify why the result noted here was surprising (page 20, beginning line 485):

“While the PRh and BLA clearly communicate to support mediated learning about the sound, this communication is not required for retrieval/expression of the mediated sound-shock association at the time of testing. This result is somewhat surprising as activity in the PRh is needed for expression of fear to the sound (Holmes et al., 2013; Wong et al., 2019) and raises the question: how does the PRh-dependent sound-shock association come to be expressed in fear responses?”

(4) The authors reference Holland 1981 and 1998, yet there's not much discussion of these findings. I think there should be a bit more emphasis on these studies since they show how mediated learning greatly depends on the extent of training. Also, it may be worth considering Holland's theory of why mediated conditioning is more effective with shorter training. His theory may be consistent with the authors, but I believe he suggests that early in training a stronger mediated representation is evoked which tends to dissipate with time. I think this is a valid hypothesis to consider in this paper.

The Holland papers show that rats form mediated associations (Holland, 1981) and that the likelihood of them doing so decreases with the amount of training (Holland, 1998). These findings are paralleled by those reported in the present series of experiments. However, the protocols used by Holland were very different to those used in the present study; and the explanation for his 1998 findings (which is the more relevant of the two papers) simply does not apply to the case of sensory preconditioning.

To be clear: Holland (1998) exposed rats to either “few” or “many” tone-food pairings in stage 1, tone-lithium chloride pairings in stage 2 and, finally, tested rats with the food alone in stage 3. He predicted and showed that those exposed to few tone-food pairings showed an aversion to the food at test (i.e., they consumed less of the food than controls) whereas those exposed to many tone-food pairings showed no such aversion (i.e., they consumed the same amount of food as the controls). This was taken to mean that, across the series of tone-lithium pairings, the tone activated the memory of food among rats in the few condition, resulting in a mediated food-lithium association; but failed to do so among rats in the many condition, resulting in no food-lithium association. According to Holland, the tone failed to activate the memory of food in the many condition because, by the end of training in stage 1, it was not needed for them to know *what to do* when the tone was presented: they simply had to run to the magazine to collect the food when delivered. That is, the tone eventually associated with the responses that rats emitted in the training situation, thereby obviating any need for activation of the food memory.

While this explanation is both elegant and interesting, it cannot be applied to the results obtained in the present study where the initial stage of training involved few or many sound-light pairings. That is, unlike in the Holland study where rats in the many condition eventually learned a stimulus-“run to magazine” association that maintained performance in the absence of any mental image of food, in the present study, any stimulus-response association acquired in stage 1 (e.g., orienting responses towards the sources of the auditory and visual stimuli) cannot have contributed to the expression of sensory preconditioned fear at test. Hence, stimulus-response learning in the many condition cannot be invoked to explain the pattern of results in the present study, even if it adequately explains what-appears-to-be a similar finding in the Holland study.

Nonetheless, we have included a reference to the general style of explanation that was considered and rejected by Holland in his 1998 and 2005 papers. This appears on page 18 (beginning line 434) and reads:

“Why does increasing the number of sound-light pairings change the way that rats integrate the sound-light and light-shock memories? One possibility is that increasing the number of sound-light pairings in stage 1 reduces the ability of each stimulus to activate the memory of the other. This is consistent with findings by Holland (1998), who showed that the likelihood of mediated learning in rats decreases with the amount of training (see also Holland, 2005); but inconsistent with our findings that, after extended training, rats continue to integrate the sound-light and light-shock associations through chaining at the time of testing (as chaining is predicated on the sound activating the memory of the light after extended training). Instead, we propose that the change in integration occurs because the increased number of sound-light pairings allows the rats to learn about the order in which the sound and light are presented (Figure 1; for evidence that rats acquire order information in sensory preconditioning, see Barnet et al., 1997; Hart et al., 2022; Leising et al., 2007; Miller & Barnet, 1993)…”

(5) There is also a Holland 2005 paper in which he tests whether extended training of the initial stimulus associations may result in a reduced associability of those stimuli. This would potentially result in lower mediated learning due to a decreased associability of the mediated representation, thereby explaining why extended training reductions in mediated learning occur. Using a probabilistic design, Holland shows that this reduction in mediated learning is likely not due to a change in associability.

We appreciate the note re Holland (2005) and have included a reference to it in our General Discussion. We agree with Holland that the reduction in mediated learning across extended training is not due to reduced associability of the retrieved stimulus representation. If this were the case, it would remain to explain why stimulus representations continue to be activated at test, which must occur for successful chaining of the sound-light and light-shock associations upon presentations of the sound alone. This is included in the modified text on page 18 (beginning line 434), which is part of our response to point 4.

**Reviewer #3 (Recommendations for the authors):**
(1) I think the 4th intro paragraph is essentially saying that more pairings during preconditioning encourage chaining as opposed to mediated learning - I might recommend clarifying this a bit. It took me a while to put it together.

Apologies for the confusion. We have clarified the argument at this point in the Introduction with the following insertion on page 4 (beginning line 84):

“That is, increasing the number of sound-light pairings may allow rats to encode information about stimulus order in stage 1 and, thereby, shift the locus of integration from mediated conditioning in stage 2 to chaining at test in stage 3 (Holmes et al., 2022).”

(2) In analyzing test data I am assuming percent freezing is the average of the entire 30s or 10s CS period - could this be clarified?

This is correct and has been clarified in the section for ‘Scoring and Statistics’ on page 29 (beginning line 708):

“Freezing data were collected using a time-sampling procedure in which each rat was scored as either ‘freezing’ or ‘not freezing’ every two seconds by an observer blind to the rat’s group allocation. A percentage score was then calculated by dividing the number of samples scored as freezing by the total number of samples. The baseline level of freezing was established by scoring the first two min at the start of each experimental session: i.e., we divided the total number of samples scored as freezing by the total number of observed samples, which was 60. The levels of freezing to the 10 s conditioned stimulus and 30 s preconditioned stimulus were established in a similar manner: we scored the entire period of each stimulus presentation and divided the number of samples scored as freezing by the total number of observed samples, which was 5 for each presentation of the conditioned stimulus and 15 for each presentation of the preconditioned stimulus.”

(3) Complementary to the above - during the probe test is there a difference during the first/last 2s of the CS? This would be interesting with respect to understanding the associative structure encoded.

We have previously examined whether freezing responses change across the duration of a 30 s preconditioned stimulus and a 10 s conditioned stimulus. We have never seen any such changes: in our past work and in the present series of experiments, the expression of freezing is largely uniform across each presentation of a preconditioned or conditioned stimulus.

(4) It is sort of unclear to me why more CS-CS pairings produced stronger preconditioned fear - is it that both mediated learning and chaining occur and giving 32 pairings permits both processes more than 8 pairings?

This is a very reasonable explanation for the heightened level of sensory preconditioned fear among rats that received many sound-light pairings in the initial control experiment. We are, however, reluctant to offer a strong interpretation of this result as it was not replicated across subsequent experiments in the series: i.e., the levels of freezing to the sensory preconditioned stimulus at test were largely the same among vehicle-injected controls that received either few (8) or many (32) sound-light pairings in Experiments 2A and 2B, and again in Experiments 3A and 3B as well as Experiments 4A and 4B.

(5) I would suggest individual data points overlaid on the bars, violin plots, or box and whisker plots to provide a better visualization of the data.

We appreciate the suggestion – these have been included overlaid on bars in each histogram.

(6) There are other citations that would strengthen arguments for the idea that unidirectional/temporal associative structure can be acquired during (appetitive) sensory preconditioning: Leising 2007 Learning and Behavior, Hart 2022 Current Biology, for example.

Thank you for these citations. We have included references to the Leising et al (2007) and Hart et al (2022) papers in our discussion on page 18-19 (beginning line 442):

“Instead, we propose that the change in integration occurs because the increased number of sound-light pairings allows the rats to learn about the order in which the sound and light are presented (Figure 1; for evidence that rats acquire order information in sensory preconditioning, see Barnet et al., 1997; Hart et al., 2022; Leising et al., 2007; Miller & Barnet, 1993)…”

Editor's note:We agree with the suggestions about full statistical reporting for non-significant results and about putting individual data points, perhaps coded to identify sex, on top of the bar graphs. Both will increase the transparency of the rigor of the work for readers.

We thank the editors and authors for their suggestions. We have included full statistical reporting for non-significant results and overlaid individual data points on the bars in each histogram.